# Learning Topology-Preserving Data Representations

**Ilya Trofimov**[1], **Daniil Cherniavskii**[1,4], **Eduard Tulchinskii**[1,3], **Nikita Balabin**[1],
**Evgeny Burnaev**[*,1,4], **Serguei Barannikov**[*,1,2]
[1]Skolkovo Institute of Science and Technology;
[2]CNRS, Université Paris Cité; [3]Huawei Noah's Ark lab;
[4]Artificial Intelligence Research Institute (AIRI)

## Abstract

We propose a method for learning topology-preserving data representations (dimensionality reduction). The method aims to provide topological similarity between the data manifold and its latent representation via enforcing the similarity in topological features (clusters, loops, 2D voids, etc.) and their localization. The core of the method is the minimization of the Representation Topology Divergence (RTD) between original high-dimensional data and low-dimensional representation in latent space. RTD minimization provides closeness in topological features with strong theoretical guarantees. We develop a scheme for RTD differentiation and apply it as a loss term for the autoencoder. The proposed method "RTD-AE" better preserves the global structure and topology of the data manifold than state-of-the-art competitors as measured by linear correlation, triplet distance ranking accuracy, and Wasserstein distance between persistence barcodes.

## 1 Introduction

Dimensionality reduction is a useful tool for data visualization, preprocessing, and exploratory data analysis. Clearly, immersion of high-dimensional data into 2D or 3D space is impossible without distortions which vary for popular methods. Dimensionality reduction methods can be broadly classified into global and local methods. Classical global methods (PCA, MDS) tend to preserve the global structure of a manifold. However, in many practical applications, produced visualizations are non-informative since they don't capture complex non-linear structures. Local methods (UMAP (McInnes et al., 2018), PaCMAP (Wang et al., 2021), t-SNE (Van der Maaten & Hinton, 2008), Laplacian Eigenmaps (Belkin & Niyogi, 2001), ISOMAP (Tenenbaum et al., 2000)) focus on preserving neighborhood data and local structure with the cost of sacrificing the global structure. The most popular methods like t-SNE and UMAP are a good choice for inferring cluster structures but often fail to describe correctly the data manifold's topology. t-SNE and UMAP have hyperparameters influencing representations neighborhood size taken into account. Different values of hyperparameters lead to significantly different visualizations and neither of them is the "canonical" one that correctly represents high-dimensional data.

We take a different perspective on dimensionality reduction. We propose the approach based on *Topological Data Analysis (TDA)*. Topological Data Analysis (Barannikov, 1994; Zomorodian, 2001; Chazal & Michel, 2017) is a field devoted to the numerical description of multi-scale topological properties of data distributions by analyzing point clouds sampled from them. TDA methods naturally capture properties of data manifolds on multiple distance scales and are arguably a good trade-off between local and global approaches.

The state-of-the-art TDA approach of this kind is TopoAE (Moor et al., 2020). However, it has several weaknesses: 1) the loss term is not continuous 2) the nullity of the loss term is only necessary but not a sufficient condition for the coincidence of topology, as measured by persistence barcodes, see more details in Appendix J.

---

*Equal senior contribution.
Correspondence: i-tr@yandex.ru

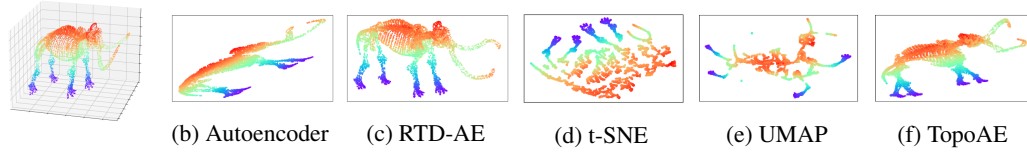

(a) Orig. 3D data  (b) Autoencoder  (c) RTD-AE  (d) t-SNE  (e) UMAP  (f) TopoAE

Figure 1: Dimensionality reduction (3D → 2D) on the "Mammoth" dataset. The proposed RTD-AE method better captures both global and local structure.

In our paper, we suggest using the Representation Topology Divergence (RTD) (Barannikov et al., 2022) to produce topology-aware dimensionality reduction. RTD measures the topological discrepancy between two point clouds with one-to-one correspondence between clouds and enjoys nice theoretical properties (Section 3.2). The major obstacle to incorporate RTD into deep learning is its differentiation. There exist approaches to the differentiation of barcodes, generic barcodes-based functions with respect to deformations of filtration (Carriére et al., 2021) and to TDA differentiation in special cases (Hofer et al., 2019; Poulenard et al., 2018).

In this paper, we make the following contributions:

1. We develop an approach for RTD differentiation. Topological metrics are difficult to differentiate; the differentiability of RTD and its implementation on GPU is a valuable step forward in the TDA context which opens novel possibilities in topological optimizations;
2. We propose a new method for topology-aware dimensionality reduction: an autoencoder enhanced with the differentiable RTD loss: "RTD-AE". Minimization of RTD loss between real and latent spaces forces closeness in topological features and their localization with strong theoretical guarantees;
3. By doing computational experiments, we show that the proposed RTD-AE outperforms state-of-the-art methods of dimensionality reduction and the vanilla autoencoder in terms of preserving the global structure and topology of a data manifold; we measure it by the linear correlation, the triplet distance ranking accuracy, Wasserstein distance between persistence barcodes, and RTD. In some cases, the proposed RTD-AE produces more faithful and visually appealing low-dimensional embeddings than state-of-the-art algorithms. We release the RTD-AE source code. [1]

## 2 RELATED WORK

Various dimensionality reduction methods have been proposed to obtain 2D/3D visualization of high-dimensional data (Tenenbaum et al., 2000; Belkin & Niyogi, 2001; Van der Maaten & Hinton, 2008; McInnes et al., 2018). Natural science researchers often use dimensionality reduction methods for exploratory data analysis or even to focus further experiments (Becht et al., 2019; Kobak & Berens, 2019; Karlov et al., 2019; Andronov et al., 2021; Szubert et al., 2019). The main problem with these methods is inevitable distortions (Chari et al., 2021; Batson et al., 2021; Wang et al., 2021) and incoherent results for different hyperparameters. These distortions can largely affect global representation structure such as inter-cluster relationships and pairwise distances. As the interpretation of these quantities in some domain such as physics or biology can lead to incorrect conclusions, it is of high importance to preserve them as much as possible. UMAP and t-SNE visualizations are frequently sporadic and cannot be considered as "canonical" representation of high-dimensional data. An often overlooked issue is the initialization which significantly contributes to the performance of dimensionality reduction methods (Kobak & Linderman, 2021; Wang et al., 2021). Damrich & Hamprecht (2021) revealed that the UMAP's true loss function is different from the purported from its theory because of negative sampling. There is a number of works that try to tackle the distortion problem and preserve as much inter-data relationships as possible. Authors of PHATE (Moon et al., 2019) and ivis (Szubert et al., 2019) claim that their methods are able to capture local as well as global features, but provide no theoretical guarantees for this. (Wagner et al., 2021)

---

[1]github.com/danchern97/RTD_AE

propose DIPOLE, an approach to dimensionality reduction combining techniques of metric geometry and distributed persistent homology.

From a broader view, deep representation learning is also dedicated to obtaining low-dimensional representation of data. Autoencoder (Hinton & Salakhutdinov, 2006) and Variational Autoencoder (Kingma & Welling, 2013) are mostly used to learn representations of objects useful for solving downstream tasks or data generation. They are not designed for data visualization and fail to preserve simultaneously local and global structure on 2D/3D spaces. Though, their parametric nature makes them scalable and applicable to large datasets, which is why they are used in methods such as parametric UMAP (Sainburg et al., 2021) and ivis (Szubert et al., 2019) and ours.

Moor et al. (2020) proposed TopoAE, including an additional loss for the autoencoder to preserve topological structures of the input space in latent representations. The topological similarity is achieved by retaining similarity in the multi-scale connectivity information. Our approach has a stronger theoretical foundation and outperforms TopoAE in computational experiments.

An approach for differentiation of persistent homology-based functions was proposed by Carriére et al. (2021). Leygonie et al. (2021) systematizes different approaches to regularisation of persistence diagrams function and defines notions of differentiability for maps to and from the space of persistence barcodes. Luo et al. (2021) proposed a topology-preserving dimensionality reduction method based on graph autoencoder. Kim et al. (2020) proposed a differentiable topological layer for general deep learning models based on persistence landscapes.

## 3 PRELIMINARIES

### 3.1 TOPOLOGICAL DATA ANALYSIS, PERSISTENT HOMOLOGY

Topology is often considered to describe the "shape of data", that is, multi-scale properties of the datasets. Topological information was generally recognized to be important for various data analysis problems. In the perspective of the commonly assumed manifold hypothesis (Goodfellow et al., 2016), datasets are concentrated near low-dimensional manifolds located in high-dimensional ambient spaces. The standard direction is to study topological features of the underlying manifold. The common approach is to cover the manifold via simplices. Given the threshold $\alpha$, we take sets of the points from the dataset $X$ which are pairwise closer than $\alpha$. The family of such sets is called the Vietoris-Rips simplicial complex. For further convenience, we introduce the fully-connected weighted graph $\mathcal{G}$ whose vertices are the points from $X$ and whose edges have weights given by the distances between the points. Then, the Vietoris-Rips simplicial complex is defined as:

$$\text{VR}_\alpha(\mathcal{G}) = \{\{i_0, \ldots, i_k\}, i_m \in \text{Vert}(\mathcal{G}) \mid m_{i,j} \leq \alpha\},$$

where $m_{i,j}$ is the distance between points, $\text{Vert}(\mathcal{G}) = \{1, \ldots, |X|\}$ is the vertices set of the graph $\mathcal{G}$.

For each $\text{VR}_\alpha(\mathcal{G})$, we define the vector space $C_k$, which consists of formal linear combinations of all $k$-dimensional simplices from $\text{VR}_\alpha(\mathcal{G})$ with modulo 2 arithmetic. The boundary operator $\partial_k : C_k \to C_{k-1}$ maps every simplex to the sum of its facets. One can show that $\partial_k \circ \partial_{k-1} = 0$ and the chain complex can be created:

$$\ldots \to C_{k+1} \overset{\partial_{k+1}}{\to} C_k \overset{\partial_k}{\to} C_{k-1} \to \ldots.$$

The quotient vector space $H_k = ker(\partial_k)/im(\partial_{k+1})$ is called the $k$-th homology group, elements of $H_k$ are called homology classes. The dimension $\beta_k = dim(H_k)$ is called the $k$-th Betti number and it approximates the number of basic topological features of the manifold represented by the point cloud $X$.

The immediate problem here is the selection of appropriate $\alpha$ which is not known beforehand. The standard solution is to analyze all $\alpha > 0$. Obviously, if $\alpha_1 \leq \alpha_2 \leq \ldots \leq \alpha_m$, then $\text{VR}_{\alpha_1}(\mathcal{G}) \subseteq \text{VR}_{\alpha_2}(\mathcal{G}) \subseteq \ldots \subseteq \text{VR}_{\alpha_m}(\mathcal{G})$; the nested sequence is called the filtration. The evolution of cycles across the nested family of simplicial complexes $S_{\alpha_i}$ is canonically decomposed into "birth" and "death" of basic topological features, so that a basic feature $c$ appears in $H_k(S_\alpha)$ at a specific threshold $\alpha_c$ and disappears at a specific threshold $\beta_c$, $\beta_c - \alpha_c$ describes the "lifespan" or persistence of the homology class. The set of the corresponding intervals $[\alpha_c, \beta_c]$ for the basic homology classes from $H_k$ is called the *persistence barcode*; the whole theory is dubbed the *persistent homology* (Chazal & Michel, 2017; Barannikov, 1994; Zomorodian, 2001).

### 3.2 REPRESENTATION TOPOLOGY DIVERGENCE (RTD)

The classic persistent homology is dedicated to the analysis of a single point cloud $X$. Recently, Representation Topology Divergence (RTD) (Barannikov et al., 2022) was proposed to measure the dissimilarity in the multi-scale topology between two point clouds $X, \tilde{X}$ of equal size $N$ with a one-to-one correspondence between clouds. Let $\mathcal{G}^w$, $\mathcal{G}^{\tilde{w}}$ be graphs with weights on edges equal to pairwise distances of $X, \tilde{X}$. To provide the comparison, the auxiliary graph $\hat{\mathcal{G}}^{w,\tilde{w}}$ with doubled set of vertices and edge weights matrix $m(w, \tilde{w})$, see details in Appendix B, is created. The persistence barcode of the graph $\hat{\mathcal{G}}^{w,\tilde{w}}$ is called the *R-Cross-Barcode* and it tracks the differences in the multi-scale topology of the two point clouds by comparing their $\alpha$-neighborhood graphs for all $\alpha$.

Here we give a simple example of an R-Cross-Barcode, see also (Cherniavskii et al., 2022). Suppose we have two point clouds $A$ and $B$, of seven points each, with distances between points as shown in the top row of Figure 2. Consider the R-Cross-Barcode$_1$(A, B), it consists of 4 intervals (the bottom row of the figure). The 4 intervals describe the topological discrepancies between connected components of $\alpha$-neighborhood graphs of $A$ and $B$.

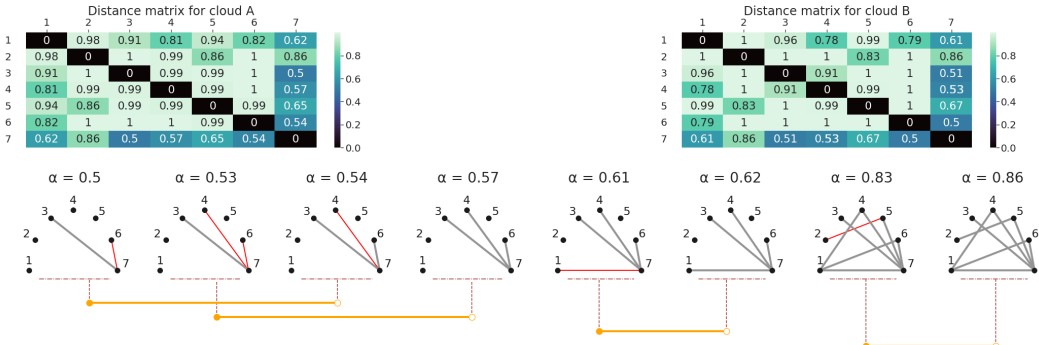

Figure 2: A graphical representation of an R-Cross-Barcode$_1(A, B)$ for the point clouds $A$ and $B$. The pairwise distance matrices for $A$ and $B$ are shown in the top raw. Edges present in the $\alpha$-neighborhood graphs for $B$ but not for $A$ are colored in red. Edges present in the $\alpha$-neighborhood graph for $A$ are colored in grey. The timeline for appearance-disappearance of topological features distinguishing the two graphs is shown. The appearance-disappearance process is illustrated by the underlying bars, connecting the corresponding thresholds.

An interval is opened, i.e. a topological discrepancy appears, at threshold $\alpha = \tilde{w}^B_{uv}$ when in the union of $\alpha$-neighborhood graph of $A$ and $B$, two vertex sets $C_1$ and $C_2$ disjoint at smaller thresholds, are joined into one connected component by the edge $(uv)$ from $B$. This interval is closed at threshold $\alpha = w^A_{u'v'}$ when the two vertex sets $C_1$ and $C_2$ are joined into one connected component in the $\alpha$-neighborhood graph of $A$.

For example, a discrepancy appears at the threshold $\alpha = 0.53$ when the vertex sets $\{4\}$ and $\{3, 6, 7\}$ are joined into one connected component in the union of neighborhood graphs of $A$ and $B$ by the edge $(4, 7)$. We identify the "death" of this R-Cross-Barcode feature at $\alpha = 0.57$, when these two sets are joined into one connected component in the neighborhood graph of cloud A (via the edge $(4, 7)$ in Figure 2 becoming grey).

By definition, RTD$_k(X, \tilde{X})$ is the sum of intervals' lengths in the *R-Cross-Barcode$_k$*$(X, \tilde{X})$ and measures its closeness to an empty set.

**Proposition 1** (Barannikov et al. (2022)). *If $RTD_k(X, \tilde{X}) = RTD_k(\tilde{X}, X) = 0$ for all $k \geq 1$, then the barcodes of the weighted graphs $\mathcal{G}^w$ and $\mathcal{G}^{\tilde{w}}$ are the same in any degree. Moreover, in this case the topological features are located in the same places: the inclusions $VR_\alpha(\mathcal{G}^w) \subseteq VR_\alpha(\mathcal{G}^{\min(w,\tilde{w})})$, $VR_\alpha(\mathcal{G}^{\tilde{w}}) \subseteq VR_\alpha(\mathcal{G}^{\min(w,\tilde{w})})$ induce homology isomorphisms for any threshold $\alpha$.*

The Proposition 1 is a strong basis for topology comparison and optimization. Given a fixed data representation $X$, how to find $\tilde{X}$ lying in a different space, and having a topology similar to $X$, in particular, similar persistence barcodes? Proposition 1 states that it is sufficient to minimize

$\sum_{i \geq 1} \left( \text{RTD}_i(X, \tilde{X}) + \text{RTD}_i(\tilde{X}, X) \right)$. In most of our experiments we minimized $\text{RTD}_1(X, \tilde{X}) + \text{RTD}_1(\tilde{X}, X)$. $\text{RTD}_1$ can be calculated faster than $\text{RTD}_{2+}$, also $\text{RTD}_{2+}$ are often close to zero. To simplify notation, we denote $\text{RTD}(X, \tilde{X}) := \frac{1}{2}(\text{RTD}_1(X, \tilde{X}) + \text{RTD}_1(\tilde{X}, X))$.

**Comparison with TopoAE loss**. TopoAE (Moor et al., 2020) is the state-of-the-art algorithm for topology-preserving dimensionality reduction. The TopoAE topological loss is based on comparison of minimum spanning trees in $X$ and $\tilde{X}$ spaces. However, it has several weak spots. First, when the TopoAE loss is zero there is no guarantee that persistence barcodes of $X$ and $\tilde{X}$ coincide. Second, the TopoAE loss can be discontinuous in rather standard situations, see Appendix J. At the same time, RTD loss is continuous, and its nullity guarantees the coincidence of persistence barcodes of $X$ and $\tilde{X}$. The continuity of the RTD loss follows from the stability of the R-Cross-Barcode$_k$ (Proposition 2).

**Proposition 2.** *(a) For any quadruple of edge weights sets $w_{ij}$, $\tilde{w}_{ij}$, $v_{ij}$, $\tilde{v}_{ij}$ on $\mathcal{G}$:*

$$d_B(\text{R-Cross-Barcode}_k(w, \tilde{w}), \text{R-Cross-Barcode}_k(v, \tilde{v})) \leq \max(\max_{ij}|v_{ij} - w_{ij}|, \max_{ij}|\tilde{v}_{ij} - \tilde{w}_{ij}|).$$

*(b) For any pair of edge weights sets $w_{ij}$, $\tilde{w}_{ij}$ on $\mathcal{G}$:*

$$\|\text{R-Cross-Barcode}_k(w, \tilde{w})\|_B \leq \max_{ij}|w_{ij} - \tilde{w}_{ij}|.$$

*(c) The expectation for the bottleneck distance between $\text{R-Cross-Barcode}_k(w, \tilde{w})$ and $\text{R-Cross-Barcode}_k(w', \tilde{w})$, where $w_{ij} = w(x_i, x_j)$, $w'_{ij} = w'(x_i, x_j)$, $\tilde{w}_{ij} = \tilde{w}(x_i, x_j)$, $w, w', \tilde{w}$ is a triple of metrics on a measure space $(\mathcal{X}, \mu)$, and $X = \{x_1, \ldots, x_n\}$, $x_i \in \mathcal{X}$ is a sample from $(\mathcal{X}, \mu)$, is upper bounded by Gromov-Wasserstein distance between $w$ and $w'$:*

$$\int_{\mathcal{X} \times \ldots \times \mathcal{X}} d_B(\text{R-Cross-Barcode}_k(w, \tilde{w}), \text{R-Cross-Barcode}_k(w', \tilde{w}))d\mu^{\otimes n} \leq n \, GW(w, w').$$

*(d) The expectation for the bottleneck norm of $\text{R-Cross-Barcode}_k(w, \tilde{w})$ for two weighted graphs with edge weights $w_{ij} = w(x_i, x_j)$, $\tilde{w}_{ij} = \tilde{w}(x_i, x_j)$, where $w, \tilde{w}$ is a pair of metrics on a measure space $(\mathcal{X}, \mu)$, and $X = \{x_1, \ldots, x_n\}$, $x_i \in \mathcal{X}$ is a sample from $(\mathcal{X}, \mu)$, is upper bounded by Gromov-Wasserstein distance between $w$ and $\tilde{w}$:*

$$\int_{\mathcal{X} \times \ldots \times \mathcal{X}} \|\text{R-Cross-Barcode}_k(w, \tilde{w})\|_B d\mu^{\otimes n} \leq n \, GW(w, \tilde{w}).$$

The proofs are given in Appendix K.

## 4 METHOD

### 4.1 DIFFERENTIATION OF RTD

We propose to use RTD as a loss in neural networks. Here we describe our approach to RTD differentiation. Denote by $\Sigma_k$ the set of all $k-$simplices in the Vietoris-Rips complex of the graph $\hat{\mathcal{G}}^{w,\tilde{w}}$, and by $\mathcal{T}_k$ the set of all intervals in the R-Cross-Barcode$_k(X, \tilde{X})$. Fix (an arbitrary) strict order on $\mathcal{T}_k$.

There exists a function $f_k : \cup_{(b_i,d_i) \in \mathcal{T}_k} \{b_i, d_i\} \to \Sigma_k$ that maps $b_i$ (or $d_i$) to a simplex $\sigma$ whose appearance leads to "birth" (or "death") of the corresponding homological class. Let

$$m_\sigma = \max_{i,j \in \sigma} m_{i,j}$$

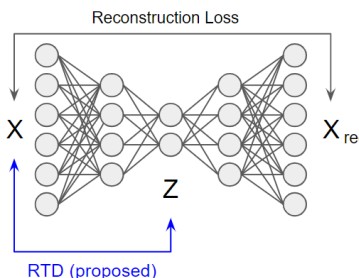

Figure 3: RTD Autoencoder

denote the function of $m_{ij}$ equal to the filtration value at which the simplex $\sigma$ joins the filtration. Since $\frac{\partial \, \text{RTD}_k(X, \tilde{X})}{\partial d_i} = -\frac{\partial \, \text{RTD}_k(X, \tilde{X})}{\partial b_i} = 1$, we obtain the following equation for the subgradient

$$\frac{\partial \, \text{RTD}_k(X, \tilde{X})}{\partial m_\sigma} = \sum_{i \in \mathcal{T}_k} \mathbb{I}\{f_k(d_i) = \sigma\} - \sum_{i \in \mathcal{T}_k} \mathbb{I}\{f_k(b_i) = \sigma\}.$$

Here, for any $\sigma$ no more than one term has non-zero indicator. Then

$$\frac{\partial \, \text{RTD}_k(X, \tilde{X})}{\partial m_{i,j}} = \sum_{\sigma \in \Sigma_k} \frac{\partial \, \text{RTD}_k(X, \tilde{X})}{\partial m_\sigma} \frac{\partial m_\sigma}{\partial m_{i,j}}.$$

The only thing that is left is to obtain subgradients of $\text{RTD}(X, \tilde{X})$ by points from $X$ and $\tilde{X}$. Consider (an arbitrary) element $m_{i,j}$ of matrix $m$. There are 4 possible scenarios:

1. $i, j \leq N$, in other words $m_{i,j}$ is from the upper-left quadrant of $m$. Its length is constant and thus $\forall l : \frac{\partial m_{i,j}}{\partial X_l} = \frac{\partial m_{i,j}}{\partial \tilde{X}_l} = 0$.

2. $i \leq N < j$, in other words $m_{i,j}$ is from the upper-right quadrant of $m$. Its length is computed as Euclidean distance and thus $\frac{\partial m_{i,j}}{\partial X_i} = \frac{X_i - X_{j-N}}{||X_i - X_{j-N}||_2}$ (similar for $X_{N-j}$).

3. $j \leq N < i$, similar to the previous case.

4. $N < i, j$, in other words $m_{i,j}$ is from the bottom-right quadrant of $m$. Here we have subgradients like

$$\frac{\partial m_{i,j}}{\partial X_{i-N}} = \frac{X_i - X_{j-N}}{||X_i - X_{j-N}||_2} \mathbb{I}\{w_{i-N,j-N} < \tilde{w}_{i-N,j-N}\}$$

Similar for $X_{j-N}, \tilde{X}_{i-N}$ and $\tilde{X}_{j-N}$.

Subgradients $\frac{\partial \text{ RTD}(X, \tilde{X})}{\partial X_i}$ and $\frac{\partial \text{ RTD}(X, \tilde{X})}{\partial \tilde{X}_i}$ can be derived from the beforementioned using the chain rule and the formula of full (sub)gradient. Now we are able to minimize $\text{RTD}(X, \tilde{X})$ by methods of (sub)gradient optimization. We discuss some possible tricks for improving RTD differentiation in Appendix I.

## 4.2 RTD Autoencoder

Given the data $X = \{x_i\}_{i=1}^n$, $x_i \in \mathbb{R}^d$, in high-dimensional space, our goal is to find the representation in low-dimensional space $Z = \{z_i\}$, $z_i \in \mathbb{R}^p$. For the visualization purposes, $p = 2, 3$. Our idea is to find a representation $Z$ which preserves *persistence barcodes*, that is, multi-scale topological properties of the point clouds, as much as possible. The straightforward approach is to solve $\min_Z \text{RTD}(X, Z)$, where the optimization is performed over $n$ vectors $z_i \in \mathbb{R}^p$, in the flavor similar to UMAP and t-SNE. This approach is workable albeit very time-consuming and could be applied only to small datasets, see Appendix F. A practical solution is to learn representations via the encoder network $E(w, x) : X \to Z$, see Figure 3.

**Algorithm**. Initially, we train the autoencoder for $E_1$ epochs with the reconstruction loss $\frac{1}{2}||X - X_{rec}||^2$ only. Then, we train for $E_2$ epochs with the loss $\frac{1}{2}||X - X_{rec}||^2 + \text{RTD}(X, Z)$. Both losses are calculated on mini-batches. The two-step procedure speedups training since calculating $\text{RTD}(X, Z)$ for the untrained network takes much time.

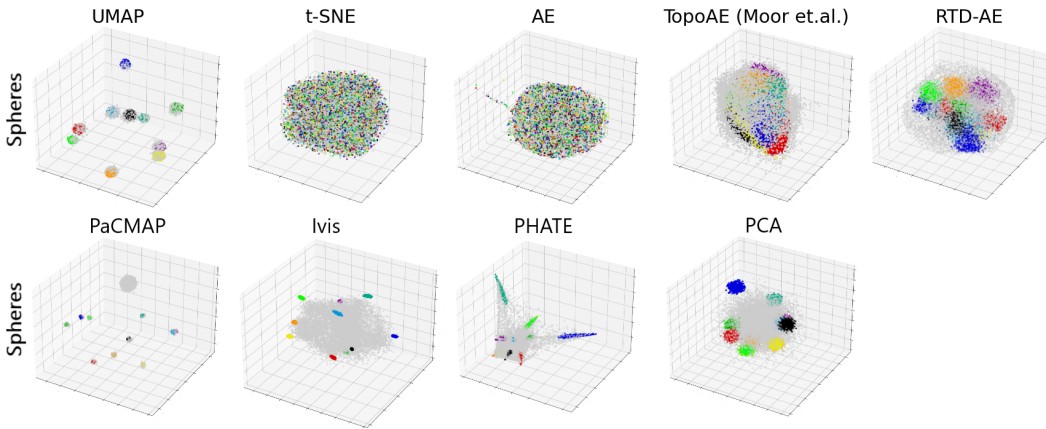

Figure 4: Results on dimensionality reduction to 3D-space

## 5 EXPERIMENTS

In computational experiments, we perform dimensionality reduction to high-dimensional and 2D/3D space for ease of visualization. We compare original data with latent representations by (1) linear correlation of pairwise distances, (2) Wasserstein distance (W.D.) between $H_0$ persistence barcodes (Chazal & Michel, 2017), (3) triplet distance ranking accuracy (Wang et al., 2021) (4) RTD. All of the quality measures are tailored to evaluate how the manifold's global structure and topology are preserved. We note that RTD, as a quality measure, provides a more precise comparison of topology than the W.D. between $H_0$ persistence barcodes. First, RTD takes into account the localization of topological features, while W.D. does not. Second, W.D. is invariant to permutations of points, but we are interested in comparison between original data and latent representation where natural one-to-one correspondence holds.

We compare the proposed RTD-AE with t-SNE (Van der Maaten & Hinton, 2008), UMAP (McInnes et al., 2018), TopoAE (Moor et al., 2020), vanilla autoencoder (AE), PHATE (Moon et al., 2019), Ivis (Szubert & Drozdov, 2019), PacMAP (Wang et al., 2021). The complete description of all the used datasets can be found in Appendix L. See hyperparameters in Appendix H.

### 5.1 SYNTHETIC DATASETS

We start with the synthetic dataset "Spheres": eleven 100D spheres in the 101D space, any two of those do not intersect and one of the spheres contains all other inside. For the visualization, we perform dimensionality reduction to 3D space. Figure 4 shows the results: RTD-AE is the best one preserving the nestedness for the "Spheres" dataset. Also, RTD-AE outperforms other methods by quality measures, see Table 1. We were unable to run MDS on "Spheres" dataset because it was too large for that method. See more results in Appendix M.

### 5.2 REAL WORLD DATASETS

We performed experiments with a number of real-world datasets: MNIST (LeCun et al., 1998), F-MNIST (Xiao et al., 2017), COIL-20 (Nene et al., 1996), scRNA mice (Yuan et al., 2017), scRNA melanoma (Tirosh et al., 2016) with latent dimension of 16 and 2, see Tables 2, 5. The choice of scRNA datasets was motivated by the increased importance of dimensionality reduction methods in natural sciences, as was previously mentioned. RTD-AE is consistently better than competitors; moreover, the gap in metrics for the latent dimension 16 is larger than such for the latent dimension 2 (see Appendix D). [2] For the latent dimension 2, RTD-AE is the first or the second one among the methods by the quality measures (see Table 5, Figure 7 in Appendix D). We conclude that the proposed RTD-AE does a good job in preserving global structure of data manifolds.

---

[2]PHATE execution take too much time and its results are no presented for many datasets.

Table 1: Quality of data manifold global structure preservation at projection from 101D into 3D space.

| Dataset | Method | Quality measure | | | |
|---|---|---|---|---|---|
| | | L. C. | W. D. $H_0$ | T. A. | RTD |
| Spheres 3D | t-SNE | 0.087 | $47.89 \pm 2.59$ | $0.206 \pm 0.01$ | $37.32 \pm 1.44$ |
| | UMAP | 0.049 | $48.31 \pm 1.83$ | $0.313 \pm 0.03$ | $44.70 \pm 1.47$ |
| | PaCMAP | 0.394 | $46.48 \pm 1.61$ | $0.156 \pm 0.02$ | $45.88 \pm 1.51$ |
| | PHATE | 0.302 | $48.78 \pm 1.65$ | $0.207 \pm 0.02$ | $44.05 \pm 1.42$ |
| | PCA | 0.155 | $47.15 \pm 1.89$ | $0.174 \pm 0.02$ | $38.96 \pm 1.25$ |
| | MDS | N.A. | N.A. | N.A. | N.A. |
| | Ivis | 0.257 | $46.32 \pm 2.04$ | $0.130 \pm 0.01$ | $41.15 \pm 1.28$ |
| | AE | 0.441 | $\mathbf{45.07 \pm 2.27}$ | $0.333 \pm 0.02$ | $39.64 \pm 1.45$ |
| | TopoAE | 0.424 | $45.89 \pm 2.35$ | $0.274 \pm 0.02$ | $38.49 \pm 1.59$ |
| | RTD-AE | $\mathbf{0.633}$ | $\mathbf{45.02 \pm 2.69}$ | $\mathbf{0.346 \pm 0.02}$ | $\mathbf{35.80 \pm 1.63}$ |

Table 2: Quality of data manifold global structure preservation at projection into 16D space.

| Dataset | Method | Quality measure | | | |
| | | L. C. | W. D. $H_0$ | T. A. | RTD |
|---|---|---|---|---|---|
| F-MNIST | UMAP | 0.602 | 592.0 ± 3.9 | 0.741 ± 0.018 | 12.31 ± 0.44 |
| | PaCMAP | 0.600 | 585.9 ± 3.2 | 0.741 ± 0.013 | 12.72 ± 0.48 |
| | Ivis | 0.582 | 552.6 ± 3.5 | 0.718 ± 0.014 | 10.76 ± 0.30 |
| | PHATE | 0.603 | 576.4 ± 4.4 | 0.756 ± 0.016 | 10.72 ± 0.15 |
| | AE | 0.879 | 320.5 ± 1.9 | 0.850 ± 0.004 | 5.52 ± 0.17 |
| | TopoAE | 0.905 | 190.7 ± 1.2 | 0.867 ± 0.006 | 3.69 ± 0.24 |
| | RTD-AE | **0.960** | **181.2 ± 0.8** | **0.907 ± 0.004** | **3.01 ± 0.13** |
| MNIST | UMAP | 0.427 | 879.1 ± 5.6 | 0.625 ± 0.016 | 17.62 ± 0.73 |
| | PaCMAP | 0.410 | 887.5 ± 6.1 | 0.644 ± 0.012 | 20.07 ± 0.70 |
| | Ivis | 0.423 | 712.6 ± 5.0 | 0.668 ± 0.013 | 12.40 ± 0.32 |
| | PHATE | 0.358 | 819.5 ± 4.0 | 0.626 ± 0.018 | 15.01 ± 0.25 |
| | AE | 0.773 | 391.0 ± 2.9 | 0.771 ± 0.010 | 7.22 ± 0.14 |
| | TopoAE | 0.801 | 367.5 ± 1.9 | 0.796 ± 0.014 | 5.84 ± 0.19 |
| | RTD-AE | **0.879** | **329.6 ± 2.6** | **0.833 ± 0.006** | **4.15 ± 0.18** |
| COIL-20 | UMAP | 0.301 | 274.7 ± 0.0 | 0.574 ± 0.011 | 15.99 ± 0.52 |
| | PaCMAP | 0.230 | 273.5 ± 0.0 | 0.548 ± 0.012 | 15.18 ± 0.35 |
| | Ivis | N.A. | N.A. | N.A. | N.A. |
| | PHATE | 0.396 | 250.7 ± 0.000 | 0.575 ± 0.014 | 13.76 ± 0.78 |
| | AE | 0.834 | 183.6 ± 0.0 | 0.809 ± 0.008 | 8.35 ± 0.15 |
| | TopoAE | 0.910 | 148.0 ± 0.0 | 0.822 ± 0.020 | 6.90 ± 0.19 |
| | RTD-AE | **0.944** | **88.9 ± 0.0** | **0.892 ± 0.007** | **5.78 ± 0.10** |
| scRNA mice | UMAP | 0.560 | 1141.0 ± 0.0 | 0.712 ± 0.010 | 21.30 ± 0.17 |
| | PaCMAP | 0.496 | 1161.3 ± 0.0 | 0.674 ± 0.016 | 21.89 ± 0.13 |
| | Ivis | 0.401 | 1082.6 ± 0.0 | 0.636 ± 0.007 | 22.56 ± 1.13 |
| | PHATE | 0.489 | 1134.6 ± 0.0 | 0.722 ± 0.013 | 21.34 ± 0.32 |
| | AE | 0.710 | 1109.2 ± 0.0 | 0.788 ± 0.013 | 20.80 ± 0.16 |
| | TopoAE | 0.634 | **826.0 ± 0.0** | 0.748 ± 0.010 | **15.37 ± 0.22** |
| | RTD-AE | **0.777** | 932.9 ± 0.0 | **0.802 ± 0.006** | 17.03 ± 0.15 |
| scRNA melanoma | UMAP | 0.474 | 1416.9 ± 9.2 | 0.682 ± 0.013 | 20.02 ± 0.35 |
| | PaCMAP | 0.357 | 1441.8 ± 9.1 | 0.681 ± 0.014 | 20.53 ± 0.36 |
| | Ivis | 0.465 | 1168.0 ± 11.4 | 0.653 ± 0.016 | 16.31 ± 0.28 |
| | PHATE | 0.427 | 1427.5 ± 9.1 | 0.687 ± 0.018 | 20.18 ± 0.41 |
| | AE | 0.458 | 1345.9 ± 11.3 | 0.708 ± 0.016 | 19.50 ± 0.37 |
| | TopoAE | 0.544 | 973.7 ± 11.1 | 0.709 ± 0.011 | 13.41 ± 0.35 |
| | RTD-AE | **0.684** | **769.5 ± 11.5** | **0.728 ± 0.017** | **10.35 ± 0.33** |

For the "Mammoth" (Coenen & Pearce, 2019b) dataset (Figure 1) we did dimensionality reduction 3D → 2D. Besides good quality measures, RTD-AE produced an appealing 2D visualization: both large-scale (shape) and low-scale (chest bones, toes, tusks) features are preserved.

## 5.3 Analysis of distortions

Next, to study distortions produced by various dimensionality reduction methods we learn transformation from 2D to 2D space, see Figure 5. Here, we observe that RTD-AE in general recovers the global structure for all of the datasets. RTD-AE typically does not suffer from the squeezing (or bottleneck) issue, unlike AE, which is noticeable in "Random", "3 Clusters" and "Circle". Whereas t-SNE and UMAP struggle to preserve cluster densities and intercluster distances, RTD-AE manages to do that in every case. It does not cluster random points together, like t-SNE. Finally, the overall shape of representations produced by RTD-AE is consistent, it does not tear apart close points, which is something UMAP does in some cases, as shown in the "Circle" dataset. The metrics, presented in the Table 6 in Appendix E, also confirm the statements above. RTD-AE has typically higher pairwise distances linear correlation and triplet accuracy, which accounts for good multi-scale properties, while having a lower Wasserstein distance between persistence barcodes.

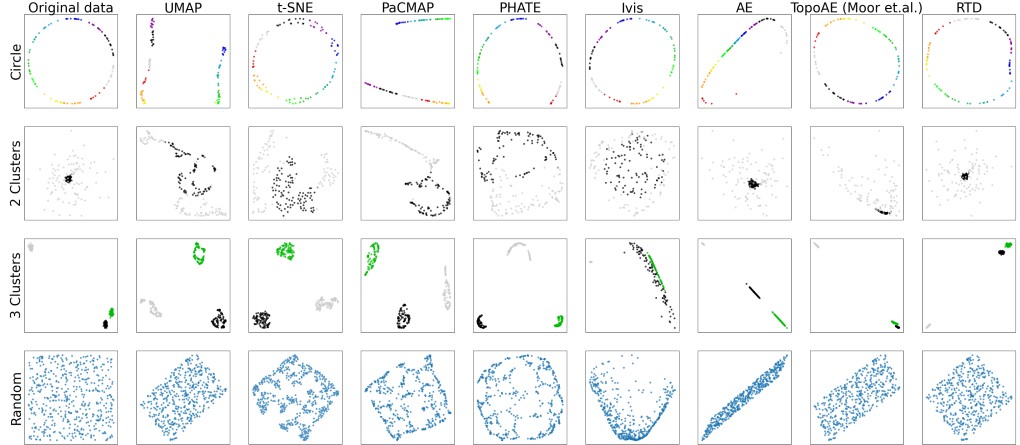

Figure 5: Results on synthetic 2D data. First column: original data. Other columns: results of dimensionality reduction methods.

## 5.4 LIMITATIONS AND COMPUTATIONAL COMPLEXITY

The main source of complexity is RTD computation. For the batch size $b$, object dimensionality $d$ and latent dimensionality $k$, the complexity is $O(b^2(d + k))$ operations since all the pairwise distances should be calculated. The R-Cross-Barcode computation is at worst cubic in the number of simplices involved. However, the computation is often quite fast for batch sizes $\leq 256$ since the boundary matrix is typically sparse for real datasets. The selection of simplices whose addition leads to "birth" or "death" of the corresponding homological class doesn't take extra time. For RTD calculation and differentiation, we used GPU-optimized software. As calculation relies heavily on the batch size, the training time of RTD-AE ranges from 1.5x the time of the basic autoencoder at batch size 8 to 4-6x the time in case of batch 512. For COIL-20, the it took ~10 minutes to train a basic AE and ~20 minutes for RTD-AE. Overall, the computation of a R-Cross-Barcode takes a similar time as in the previous step even on datasets of big dimensionality.

## 5.5 DISCUSSION

Experimental results show that RTD-AE better preserves the data manifold global structure than its competitors. The most interesting comparison is with TopoAE, the state-of-the-art, which uses an alternative topology-preserving loss. The measures of interest for topology comparison are the Wasserstein distances between persistence barcodes. Tables 2, 6, 5 show that RTD-AE is better than TopoAE. RTD minimization has a stronger theoretical foundation than the loss from TopoAE (see Section 3.2).

## 6 CONCLUSIONS

In this paper, we have proposed an approach for topology-preserving representation learning (dimensionality reduction). The topological similarity between data points in original and latent spaces is achieved by minimizing the Representation Topology Divergence (RTD) between original data and latent representations. Our approach is theoretically sound: RTD=0 means that persistence barcodes of any degree coincide and the topological features are located in the same places. We proposed how to make RTD differentiable and implemented it as an additional loss to the autoencoder, constructing RTD-autoencoder (RTD-AE). Computational experiments show that the proposed RTD-AE better preserves the global structure of the data manifold (as measured by linear correlation, triplet distance ranking accuracy, Wasserstein distance between persistence barcodes) than popular methods t-SNE and UMAP. Also, we achieve higher topological similarity than the alternative TopoAE method. Of course, the application of RTD loss is not limited to autoencoders and we expect more deep learning applications involving one-to-one correspondence between points. The main limitation is that calculation of persistence barcodes and RTD, in particular, is computationally demanding. We see here another opportunity for further research.

## ACKNOWLEDGEMENTS

The work was supported by the Analytical center under the RF Government (subsidy agreement 000000D730321P5Q0002, Grant No. 70-2021-00145 02.11.2021)

## REPRODUCIBILITY STATEMENT

To provide reproducibility, we release the source code of the proposed RTD-AE, see section 1, for hyperparameters see Appendix H. For other methods, we used either official implementations or implementations from scikit-learn with default hyperparameters. We used public datasets (see Section 5, Appendix L). We generated several synthetic datasets and made the generating code available.

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

## A  SIMPLICIAL COMPLEXES AND FILTRATIONS

Here we briefly recall basic topological objects mentioned in our paper. Suppose we have a full graph $X = \{x_0, x_1, \dots x_n\}$ a set of points in some metric space $(R, d)$.

**Definition A.1** Any set $\sigma \subseteq X$ is *(combinatorial) simplex*. Its *vertices* are all points that belong to $K$. Its *dimensionality* is number equal to $|\sigma| - 1$. Its *faces* are all proper subsets of $\sigma$.

**Definition A.2** *Simplicial complex* $\mathcal{C}$ is a set of simplices such that for every simplex $\sigma \in \mathcal{C}$ it contains all faces of $\sigma$ and for every two simplices $\sigma_1, \sigma_2 \in \mathcal{C}$ their intersection is face of both of them.

Simplicial complexes can be seen as higher-dimensional generalization of graphs. There are many ways to build a simplicial complex from a set of points, but only two important for this work: Vietoris-Rips and Chech complexes.

**Definition A.3** Given a threshold $\alpha$ the *Vietoris-Rips (simplicial) complex* at threshold $\alpha$ (denotes as $\text{VR}_\alpha(X)$) is defined as set set of all simplices $\sigma$ such that $\forall x_i, x_j \in \sigma$ holds $d(x_i, x_j) \leq \alpha$.

**Definition A.4** Given a threshold $\alpha$ the *Čech (simplicial) complex* at threshold $\alpha$ (denotes as $\text{Cech}_\alpha(X)$) is defined as set set of all simplices $\sigma$ such that all closed balls of radius $\alpha$ and with centers in vertices of $\sigma$ have a non-empty intersection.

Alhough Čech complexes are rarely used in applications of Topological Data Analysis, they are important due to the fact that their fundamental topological properties are equal to those of the manifold 'behind' X (so-called *Nerve theorem*, see (Chazal & Michel, 2017) for proper explanation).

The Vietoris-Rips complexes 'approximate' Čech complexes :

$$\text{VR}_\alpha(X) \subseteq \text{Cech}_\alpha(X) \subseteq \text{VR}_{2\alpha}(X)$$

Note that the definition of the Vietoris-Rips complex doesn't require (even indirectly) function $d(.)$ to be metric - it should only be symmetric and non-negative. And so we can define the Vietoris-Rips complex of a weighted graph $G = (V, E)$. To do so we modify Definition A.3 by replacing X with $V$ and taking $d(v_i, v_j)$ as the weight of the edge between $v_i$ and $v_j$ for $i \neq j$ and $d(v_i, v_i) = 0, \forall i$.

In the scope of this work we consider only Vietoris-Rips complexes of graphs.

**Definition A.5** A *filtration* of a simplicial complex $\mathcal{C}$ is a nested family of subcomplexes $(\mathcal{C}_t)_{t \in T}$, where $T \subseteq R$, such that for any $t_1, t_2 \in T$, if $t_1 \leq t_2$ then $\mathcal{C}_{t_1} \subseteq \mathcal{C}_{t_2}$, and $\mathcal{C} = \bigcup_{t \in T} \mathcal{C}_t$. The set T may be either finite or infinite.

Vietoris-Rips filtration can 'reflect' topology of data set at every scale. Usually data sets are finite so there is finite number of thresholds that give different Vietoris-Rips complexes and thus finite filtration is enough for it.

## B  FORMAL DEFINITION OF RTD

The classic persistent homology is dedicated to the analysis of a single point cloud $X$. Recently, Representation Topology Divergence (RTD) (Barannikov et al., 2022) was proposed to measure the dissimilarity in the multi-scale topology between two point clouds $X, \tilde{X}$ of equal size $N$ with a one-to-one correspondence between clouds.

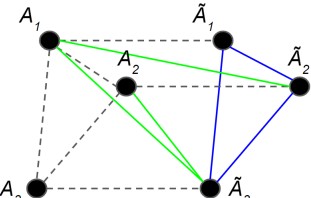

Let $\text{VR}_\alpha(\mathcal{G}^w)$, $\text{VR}_\alpha(\mathcal{G}^{\tilde{w}})$ be two Vietoris-Rips simplicial complexes, where $w, \tilde{w}$ - are the distance matrices of $X, \tilde{X}$. The idea behind RTD is to compare $\text{VR}_\alpha(\mathcal{G}^w)$ with $\text{VR}_\alpha(\mathcal{G}^{min(w,\tilde{w})})$, where $\mathcal{G}^{min(w,\tilde{w})}$ is the graph having weights $min(w, \tilde{w})$ on its edges. By definition, $\text{VR}_\alpha(\mathcal{G}^w) \subseteq \text{VR}_\alpha(\mathcal{G}^{min(w,\tilde{w})})$, $\text{VR}_\alpha(\mathcal{G}^{\tilde{w}}) \subseteq \text{VR}_\alpha(\mathcal{G}^{min(w,\tilde{w})})$.

To compare $\text{VR}_\alpha(\mathcal{G}^w)$ with $\text{VR}_\alpha(\mathcal{G}^{min(w,\tilde{w})})$, the auxiliary graph is constructed with doubled set of vertices $\hat{\mathcal{G}}^{w,\tilde{w}}$ (Figure 6) and weights

Figure 6: The graph $\hat{\mathcal{G}}^{w,\tilde{w}}$ to compare $\mathcal{G}^w = \{A_1, A_2, A_3\}$ and $\mathcal{G}^{\tilde{w}} = \{\tilde{A}_1, \tilde{A}_2, \tilde{A}_3\}$. Dashed edges correspond to zero weights, green edges to $w$, blue edges to $min(w, \tilde{w})$; edges with weight $+\infty$ are not shown.

on edges given in the simplest case by:

$$m = \begin{pmatrix} 0 & (w_+)^\mathsf{T} \\ w_+ & \min(w, \tilde{w}) \end{pmatrix},$$

where $w_+$ is the $w$ matrix with lower-triangular part replaced by $+\infty$, see ((Barannikov et al., 2022), section 2.2) for the general form of the matrix $m$. The persistence barcode of the weighted graph $\mathrm{VR}(\hat{\mathcal{G}}^{w,\tilde{w}})$ is called the *R-Cross-Barcode* (for *Representations' Cross-Barcode*). Note that for every two nodes in the graph $\hat{\mathcal{G}}^{w,\tilde{w}}$ there exists a path with edges having zero weights. Thus, the $H_0$ barcode in the *R-Cross-Barcode* is always empty.

Intuitively, the $k-$th barcode of $\mathrm{VR}_\alpha(\hat{\mathcal{G}}^{w,\tilde{w}})$ records the $k$-dimensional topological features that are born in $\mathrm{VR}_\alpha(\mathcal{G}^{\min(w,\tilde{w})})$ but are not yet born near the same place in $\mathrm{VR}_\alpha(\mathcal{G}^w)$, and the $(k-1)-$dimensional topological features that are dead in $\mathrm{VR}_\alpha(\mathcal{G}^{\min(w,\tilde{w})})$ but are not yet dead in $\mathrm{VR}_\alpha(\mathcal{G}^w)$. The *R-Cross-Barcode*$_k(X, \tilde{X})$ records the differences in the multi-scale topology of the two point clouds. The topological features with longer lifespans indicate in general the essential features.

Basic properties of R-Cross-Barcode$_k(X, \tilde{X})$ (Barannikov et al. (2022)) are:

- if $X = \tilde{X}$, then for all $k$ R-Cross-Barcode$_k(X, \tilde{X}) = \varnothing$;

- if all distances within $\tilde{X}$ are zero i.e. all objects are represented by the same point in $\tilde{X}$, then for all $k \geq 0$: R-Cross-Barcode$_{k+1}(X, \tilde{X}) = $ Barcode$_k(X)$ the standard barcode of the point cloud $X$;

- for any value of threshold $\alpha$, the following sequence of natural linear maps of homology groups

$$\xrightarrow{r_{3i+3}} H_i(VR_\alpha(\mathcal{G}^w)) \xrightarrow{r_{3i+2}} H_i(VR_\alpha(\mathcal{G}^{\min(w,\tilde{w})})) \xrightarrow{r_{3i+1}}$$
$$\xrightarrow{r_{3i+1}} H_i(VR_\alpha(\hat{\mathcal{G}}^{w,\tilde{w}})) \xrightarrow{r_{3i}} H_{i-1}(VR_\alpha(\mathcal{G}^w)) \xrightarrow{r_{3i-1}}$$
$$\xrightarrow{r_{3i-1}} \ldots \xrightarrow{r_1} H_0(VR_\alpha(\mathcal{G}^{\min(w,\tilde{w})})) \xrightarrow{r_0} 0 \quad (1)$$

is exact, i.e. for any $j$ the kernel of the map $r_j$ is the image of the map $r_{j+1}$.

**Proposition 3.** *Given an exact sequence as in (1) with finite-dimensional filtered complexes $A_\alpha$, $B_\alpha$, $C_\alpha$, the alternating sums over $k$ of their topological features lifespans satisfy*

$$\sum_k (-1)^k l_k(A) - \sum_k (-1)^k l_k(B) + \sum_k (-1)^k l_k(C) = 0 \quad (2)$$

*where $l_k(Z)$ denotes the sum of bars lengths in Barcode$_k(Z)$, here for simplicity all lifespans are assumed to be finite.*

*Proof.* The exact sequence implies that the alternating sums of dimensions of the homology groups satisfy, for any $\alpha$,

$$\sum_k (-1)^k \dim H_k(A_\alpha) - \sum_k (-1)^k \dim H_k(B_\alpha) + \sum_k (-1)^k \dim H_k(C_\alpha) = 0$$

Notice that for any $\alpha_1 < \alpha_2$

$$\dim H_k(Z_{\alpha_2}) - \dim H_k(Z_{\alpha_1}) = \#b(Z, (\alpha_1, \alpha_2], k) - \#d(Z, (\alpha_1, \alpha_2], k)$$

where $\#b(Z, (\alpha_1, \alpha_2], k)$, respectfully $\#d(Z, (\alpha_1, \alpha_2], k)$, is the number of births, respectfully deaths, of dimension $k$ topological features in $Z$ at thresholds $\alpha$, $\alpha_1 < \alpha \leq \alpha_2$. Hence

$$\sum_k (-1)^k (\#b - \#d)(A, (\alpha_1, \alpha_2], k) - \sum_k (-1)^k (\#b - \#d)(B, (\alpha_1, \alpha_2], k) +$$
$$+ \sum_k (-1)^k (\#b - \#d)(C, (\alpha_1, \alpha_2], k) = 0 \quad (3)$$

Table 3: Quality of data manifold global structure preservation at projection into 16D space.

| Dataset | Method | Quality measure | | |
|---------|--------|------|------|------|
| | | L. C. | W. D. $H_0$ | T. A. |
| F-MNIST | PCA | **0.977** | $351.3 \pm 1.7$ | $\mathbf{0.951 \pm 0.005}$ |
| | RTD-AE | 0.960 | $\mathbf{181.2 \pm 0.8}$ | $0.907 \pm 0.004$ |
| MNIST | PCA | **0.911** | $397.4 \pm 1.3$ | $\mathbf{0.863 \pm 0.010}$ |
| | RTD-AE | 0.879 | $\mathbf{329.6 \pm 2.6}$ | $0.833 \pm 0.006$ |
| COIL-20 | PCA | **0.966** | $196.4 \pm 0.0$ | $\mathbf{0.933 \pm 0.004}$ |
| | RTD-AE | 0.944 | $\mathbf{88.9 \pm 0.0}$ | $0.892 \pm 0.007$ |

Setting $\alpha_1 = \alpha - \epsilon$, $\alpha_2 = \alpha + \epsilon$, we get, for any $\alpha$

$$\sum_k (-1)^k (\#b - \#d)(A, \alpha, k) - \sum_k (-1)^k (\#b - \#d)(B, \alpha, k) + \sum_k (-1)^k (\#b - \#d)(C, \alpha, k) = 0$$

where $\#b(Z, \alpha, k)$, respectfully $\#d(Z, \alpha, k)$, is the number of births, respectfully deaths, of dimension $k$ topological features in $Z$ at the threshold $\alpha$. Summing this over all nontrivial filtration steps $\alpha$ gives the identity (2). □

**Proposition 4.**

$$\sum_k (-1)^k RTD_k(w, \tilde{w}) - \sum_k (-1)^k l_k(VR(\mathcal{G}^w)) + \sum_k (-1)^k l_k(VR(\mathcal{G}^{\min(w, \tilde{w})}))) = 0 \quad (4)$$

## C  LEARNING REPRESENTATIONS IN HIGHER DIMENSIONS

The following table shows results of the experiment with latent dimensions 16, 32, 64 and 128 for the F-MNIST dataset. RTD-AE are consistently better than the competitors.

## D  REAL WORLD DATASETS, 2D LATENT SPACE

Table 5 and Figure 7 present the results.

## E  SYNTHETIC DATASETS, 2D LATENT SPACE

Table 6 shows the results.

## F  RTD MINIMIZATION WITHOUT THE AUTOENCODER

Given the set $X = \{x_i\}_{i=1}^n$ of $n$ objects in high-dimensional space $x_i \in \mathbb{R}^d$, our goal is to find their representations in low-dimensional space $Z = \{z_i\}$, $z_i \in \mathbb{R}^k$. It is possible to solve

$$\min_Z RTD(X, Z)$$

directly w.r.t $n$ vectors $z_i \in \mathbb{R}^k$, in the flavor similar to UMAP and t-SNE. Figures 8, 9 show the results of two experiments with 3D→2D dimensionality reduction. We conclude that dimensionality reduction via RTD optimization better preserves data topology: meridians are kept connected (Figure 8) and the nestedness is retained (Figure 9). The optimization took ∼1 hour. For the experiment with nested spheres, the RTD optimization was warmstarted with the MDS solution.

Table 4: Quality of data manifold global structure preservation at projection into high-dimensional space.

| Dataset | Method | Quality measure | | | | |
|---|---|---|---|---|---|---|
| | | L. C. | W. D. $H_0$ | W. D. $H_1$ | T. A. | RTD |
| F-MNIST-128D | UMAP | 0.605 | $594.2 \pm 3.0$ | $20.73 \pm 0.56$ | $0.739 \pm 0.010$ | $12.33 \pm 0.22$ |
| | PCA | **0.996** | $107.4 \pm 2.0$ | $11.46 \pm 0.39$ | $\mathbf{0.981 \pm 0.002}$ | $1.93 \pm 0.08$ |
| | PaCMAP | 0.589 | $587.9 \pm 5.4$ | $20.90 \pm 0.27$ | $0.736 \pm 0.012$ | $12.94 \pm 0.51$ |
| | Ivis | 0.521 | $551.7 \pm 5.6$ | $23.99 \pm 0.69$ | $0.693 \pm 0.011$ | $10.46 \pm 0.23$ |
| | PHATE | 0.604 | $577.7 \pm 3.8$ | $21.57 \pm 0.52$ | $0.753 \pm 0.010$ | $10.61 \pm 0.24$ |
| | AE | 0.892 | $240.5 \pm 3.2$ | $22.88 \pm 1.14$ | $0.860 \pm 0.006$ | $4.34 \pm 0.15$ |
| | TopoAE | 0.954 | $66.2 \pm 2.1$ | $12.02 \pm 0.65$ | $\underline{0.902 \pm 0.005}$ | $2.16 \pm 0.14$ |
| | RTD-AE | 0.943 | $\mathbf{16.9 \pm 1.8}$ | $\mathbf{9.43 \pm 0.73}$ | $0.884 \pm 0.010$ | $\mathbf{1.41 \pm 0.09}$ |
| F-MNIST-64D | UMAP | 0.596 | $590.7 \pm 4.3$ | $20.27 \pm 0.71$ | $0.735 \pm 0.021$ | $12.38 \pm 0.35$ |
| | PCA | **0.992** | $179.1 \pm 1.9$ | $18.61 \pm 0.44$ | $\mathbf{0.970 \pm 0.003}$ | $3.10 \pm 0.09$ |
| | PaCMAP | 0.510 | $590.5 \pm 3.4$ | $21.37 \pm 0.47$ | $0.731 \pm 0.014$ | $13.10 \pm 0.37$ |
| | Ivis | 0.521 | $537.6 \pm 3.3$ | $26.86 \pm 0.51$ | $0.691 \pm 0.011$ | $10.34 \pm 0.31$ |
| | PHATE | 0.586 | $586.1 \pm 3.2$ | $20.78 \pm 0.52$ | $0.751 \pm 0.012$ | $10.67 \pm 0.36$ |
| | AE | 0.888 | $281.0 \pm 2.2$ | $24.78 \pm 0.86$ | $0.861 \pm 0.007$ | $4.85 \pm 0.18$ |
| | TopoAE | 0.938 | $89.3 \pm 1.8$ | $15.27 \pm 0.68$ | $0.889 \pm 0.005$ | $2.56 \pm 0.13$ |
| | RTD-AE | 0.954 | $\mathbf{57.0 \pm 0.6}$ | $\mathbf{11.76 \pm 0.28}$ | $\underline{0.895 \pm 0.008}$ | $\mathbf{1.48 \pm 0.09}$ |
| F-MNIST-32D | UMAP | 0.593 | $597.1 \pm 5.3$ | $20.39 \pm 0.24$ | $0.741 \pm 0.013$ | $12.11 \pm 0.30$ |
| | PCA | **0.986** | $263.0 \pm 2.3$ | $24.76 \pm 0.97$ | $\mathbf{0.960 \pm 0.006}$ | $4.47 \pm 0.12$ |
| | PaCMAP | 0.585 | $589.1 \pm 4.9$ | $21.15 \pm 0.55$ | $0.738 \pm 0.010$ | $12.61 \pm 0.36$ |
| | Ivis | 0.696 | $559.8 \pm 4.0$ | $23.80 \pm 0.57$ | $0.770 \pm 0.014$ | $10.14 \pm 0.29$ |
| | PHATE | 0.599 | $576.7 \pm 3.5$ | $21.79 \pm 0.69$ | $0.753 \pm 0.011$ | $10.48 \pm 0.24$ |
| | AE | 0.904 | $302.2 \pm 2.6$ | $26.37 \pm 0.74$ | $0.870 \pm 0.008$ | $5.28 \pm 0.17$ |
| | TopoAE | 0.942 | $120.9 \pm 2.5$ | $15.84 \pm 0.57$ | $0.892 \pm 0.006$ | $2.49 \pm 0.10$ |
| | RTD-AE | 0.963 | $\mathbf{108.7 \pm 1.8}$ | $\mathbf{14.03 \pm 0.90}$ | $\underline{0.907 \pm 0.006}$ | $\mathbf{1.85 \pm 0.06}$ |
| F-MNIST-16D | UMAP | 0.588 | $592.2 \pm 4.0$ | $\mathbf{20.37 \pm 0.37}$ | $0.739 \pm 0.013$ | $12.31 \pm 0.44$ |
| | PCA | **0.977** | $351.3 \pm 1.7$ | $29.15 \pm 1.08$ | $\mathbf{0.951 \pm 0.005}$ | $5.91 \pm 0.19$ |
| | PaCMAP | 0.600 | $585.9 \pm 3.2$ | $21.94 \pm 0.59$ | $0.741 \pm 0.013$ | $12.72 \pm 0.48$ |
| | Ivis | 0.582 | $552.6 \pm 3.5$ | $24.83 \pm 0.53$ | $0.718 \pm 0.014$ | $10.76 \pm 0.30$ |
| | PHATE | 0.603 | $576.4 \pm 4.4$ | $21.61 \pm 0.52$ | $0.756 \pm 0.016$ | $10.72 \pm 0.15$ |
| | AE | 0.879 | $320.5 \pm 1.9$ | $27.01 \pm 0.89$ | $0.850 \pm 0.004$ | $5.52 \pm 0.17$ |
| | TopoAE | 0.905 | $190.7 \pm 1.2$ | $25.65 \pm 1.06$ | $0.867 \pm 0.006$ | $3.69 \pm 0.24$ |
| | RTD-AE | $\underline{0.960}$ | $\mathbf{181.2 \pm 0.8}$ | $\mathbf{20.94 \pm 0.80}$ | $\underline{0.907 \pm 0.004}$ | $\mathbf{3.01 \pm 0.13}$ |

## G  ALTERNATIVE RTD VARIANT

RTD relies on the auxiliary graph with doubled set of vertices $\hat{\mathcal{G}}^{w,\tilde{w}}$ and weights on edges:

$$m = \begin{pmatrix} 0 & (w_+)^{\mathsf{T}} \\ w_+ & \min(w, \tilde{w}) \end{pmatrix}.$$

An alternative variant of RTD is possible with the following matrix of weights:

$$m = \begin{pmatrix} 0 & \max(w, \tilde{w})_+^{\mathsf{T}} \\ \max(w, \tilde{w})_+ & w \end{pmatrix}$$

in the simplest case. Both of them share similar properties and guarantee that $\text{RTD}(X, Z) = 0$ when all the pairwise distances in point clouds $X$ and $Z$ are the same. Also in both cases if $RTD_k(X, Z) = RTD_k(Z, X) = 0$ for $k \geq 1$ then the persistence diagrams of $X$ and $Z$ coincide. The minimization of the sum of both variants of RTD leads to richer gradient information. We used this loss in the experiment with the "Mammoth" dataset.

Table 5: Quality of data manifold global structure preservation for real-world data dimension reduction to 2D.

| Dataset | Method | Quality measure | | | |
| --- | --- | --- | --- | --- | --- |
| | | L. C. | W. D. $H_0$ | T. A. | RTD |
| Mammoth | t-SNE | 0.787 | $21.31 \pm 0.25$ | $0.830 \pm 0.011$ | $5.52 \pm 0.12$ |
| | UMAP | 0.776 | $28.64 \pm 0.25$ | $0.801 \pm 0.016$ | $6.81 \pm 0.25$ |
| | AE | 0.966 | $21.94 \pm 0.25$ | $0.935 \pm 0.005$ | $6.38 \pm 0.22$ |
| | PaCMAP | 0.868 | $21.13 \pm 0.21$ | $0.866 \pm 0.008$ | $5.91 \pm 0.29$ |
| | Ivis | 0.737 | $\mathbf{13.48 \pm 0.30}$ | $0.764 \pm 0.007$ | $6.14 \pm 0.20$ |
| | TopoAE | 0.915 | $21.51 \pm 0.22$ | $0.886 \pm 0.007$ | $5.16 \pm 0.08$ |
| | RTD-AE | **0.972** | $17.45 \pm 0.23$ | $\mathbf{0.928 \pm 0.006}$ | $\mathbf{3.87 \pm 0.07}$ |
| F-MNIST | t-SNE | 0.547 | $602.9 \pm 2.8$ | $0.695 \pm 0.011$ | $11.11 \pm 0.28$ |
| | UMAP | 0.595 | $616.5 \pm 2.8$ | $0.722 \pm 0.011$ | $11.72 \pm 0.24$ |
| | AE | 0.762 | $614.7 \pm 3.1$ | $0.736 \pm 0.012$ | $11.51 \pm 0.42$ |
| | PaCMAP | 0.630 | $612.8 \pm 6.0$ | $0.732 \pm 0.010$ | $11.48 \pm 0.27$ |
| | Ivis | 0.496 | $609.2 \pm 5.8$ | $0.694 \pm 0.011$ | $11.70 \pm 0.29$ |
| | PHATE | 0.613 | $608.2 \pm 2.7$ | $0.739 \pm 0.012$ | $11.60 \pm 0.22$ |
| | TopoAE | **0.795** | $\mathbf{599.0 \pm 2.9}$ | $\mathbf{0.827 \pm 0.011}$ | $11.84 \pm 0.43$ |
| | RTD-AE | 0.789 | $\mathbf{600.0 \pm 3.1}$ | $0.807 \pm 0.011$ | $\mathbf{10.67 \pm 0.26}$ |
| MNIST | t-SNE | 0.355 | $890.1 \pm 6.8$ | $0.611 \pm 0.016$ | $\mathbf{15.89 \pm 0.28}$ |
| | UMAP | 0.347 | $905.5 \pm 6.7$ | $0.612 \pm 0.020$ | $16.49 \pm 0.21$ |
| | AE | 0.415 | $892.7 \pm 5.9$ | $0.635 \pm 0.011$ | $16.41 \pm 0.27$ |
| | PaCMAP | 0.310 | $902.8 \pm 7.0$ | $0.596 \pm 0.015$ | $16.41 \pm 0.27$ |
| | Ivis | 0.377 | $\mathbf{887.5 \pm 6.8}$ | $0.630 \pm 0.014$ | $16.03 \pm 0.24$ |
| | PHATE | 0.389 | $899.7 \pm 3.4$ | $0.623 \pm 0.013$ | $16.21 \pm 0.29$ |
| | TopoAE | 0.349 | $891.5 \pm 4.6$ | $0.612 \pm 0.014$ | $\mathbf{15.71 \pm 0.09}$ |
| | RTD-AE | **0.501** | $\mathbf{885.1 \pm 4.9}$ | $\mathbf{0.664 \pm 0.009}$ | $\mathbf{15.79 \pm 0.38}$ |
| COIL-20 | t-SNE | 0.462 | $273.9 \pm 0.0$ | $0.648 \pm 0.025$ | $12.23 \pm 0.27$ |
| | UMAP | 0.247 | $279.5 \pm 0.0$ | $0.587 \pm 0.013$ | $13.72 \pm 0.35$ |
| | AE | 0.667 | $271.8 \pm 0.0$ | $0.750 \pm 0.016$ | $11.82 \pm 0.28$ |
| | PaCMAP | 0.506 | $276.6 \pm 0.0$ | $0.670 \pm 0.012$ | $12.60 \pm 0.42$ |
| | Ivis | N.A. | N.A. | N.A. | N.A. |
| | PHATE | 0.305 | $272.4 \pm 0.000$ | $0.592 \pm 0.018$ | $13.11 \pm 0.39$ |
| | TopoAE | 0.465 | $\mathbf{261.4 \pm 0.0}$ | $0.662 \pm 0.013$ | $12.18 \pm 0.23$ |
| | RTD-AE | **0.769** | $\underline{262.9 \pm 0.0}$ | $\mathbf{0.796 \pm 0.009}$ | $\mathbf{11.51 \pm 0.19}$ |
| scRNA mice | t-SNE | 0.634 | $1151.3 \pm 0.0$ | $0.749 \pm 0.010$ | $21.93 \pm 0.10$ |
| | UMAP | 0.513 | $1161.3 \pm 0.0$ | $0.709 \pm 0.015$ | $22.26 \pm 0.09$ |
| | PCA | 0.733 | $1147.3 \pm 0.0$ | $0.790 \pm 0.015$ | $22.05 \pm 0.06$ |
| | AE | 0.677 | $\mathbf{1142.2 \pm 0.0}$ | $0.778 \pm 0.008$ | $21.34 \pm 0.17$ |
| | PaCMAP | 0.483 | $1167.1 \pm 0.0$ | $0.693 \pm 0.015$ | $22.44 \pm 0.08$ |
| | Ivis | 0.254 | $1146.8 \pm 0.0$ | $0.602 \pm 0.009$ | $22.49 \pm 0.25$ |
| | PHATE | 0.522 | $1159.0 \pm 0.0$ | $0.711 \pm 0.021$ | $22.09 \pm 0.07$ |
| | TopoAE | 0.628 | $1144.2 \pm 0.0$ | $0.753 \pm 0.019$ | $\mathbf{21.15 \pm 0.10}$ |
| | RTD-AE | **0.780** | $\mathbf{1142.3 \pm 0.0}$ | $\mathbf{0.797 \pm 0.010}$ | $\mathbf{21.03 \pm 0.13}$ |
| scRNA melanoma | t-SNE | 0.505 | $1445.6 \pm 3.2$ | $0.699 \pm 0.019$ | $20.45 \pm 0.38$ |
| | UMAP | 0.471 | $1459.5 \pm 3.0$ | $0.684 \pm 0.014$ | $20.87 \pm 0.43$ |
| | PCA | 0.536 | $1446.7 \pm 3.1$ | $0.722 \pm 0.014$ | $20.64 \pm 0.39$ |
| | AE | 0.407 | $1442.0 \pm 3.5$ | $0.684 \pm 0.014$ | $20.90 \pm 0.41$ |
| | PaCMAP | 0.401 | $1460.9 \pm 3.1$ | $0.674 \pm 0.017$ | $20.91 \pm 0.38$ |
| | Ivis | 0.504 | $1442.5 \pm 3.1$ | $0.699 \pm 0.016$ | $20.50 \pm 0.35$ |
| | PHATE | 0.427 | $1458.8 \pm 3.1$ | $0.689 \pm 0.023$ | $20.94 \pm 0.41$ |
| | TopoAE | 0.521 | $1442.9 \pm 3.2$ | $0.709 \pm 0.009$ | $20.23 \pm 0.38$ |
| | RTD-AE | **0.639** | $\mathbf{1438.2 \pm 3.0}$ | $\mathbf{0.747 \pm 0.009}$ | $\mathbf{19.81 \pm 0.37}$ |

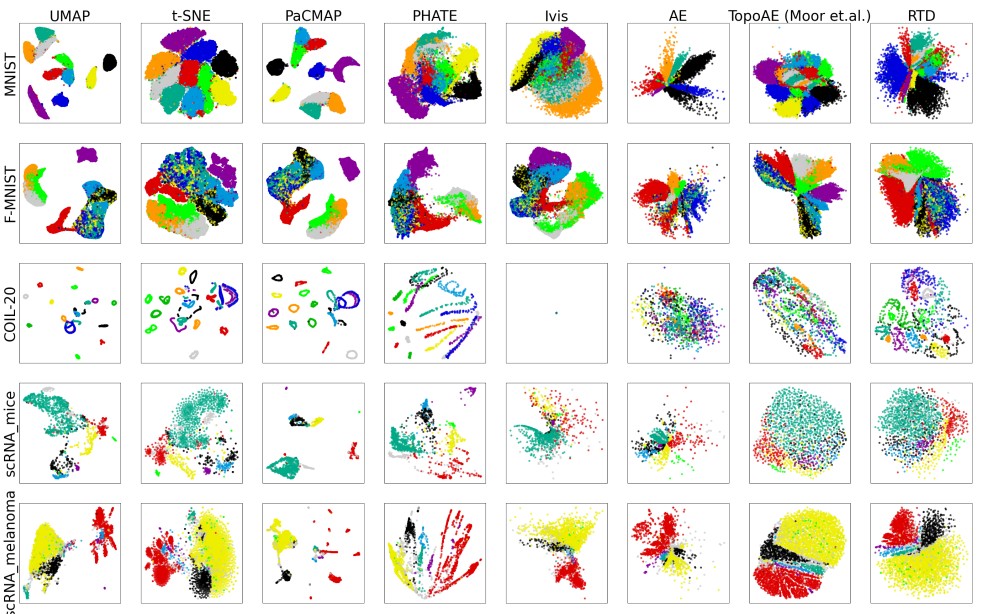

Figure 7: Results on real-world data reduction to 2D.

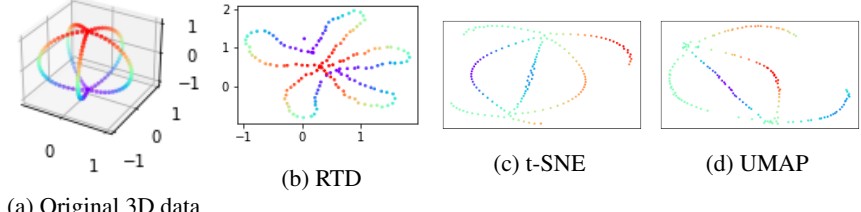

(a) Original 3D data     (b) RTD     (c) t-SNE     (d) UMAP

Figure 8: "Meridians on the sphere" dataset. Notice the disconnectedness of meridians in (c) and (d).

# H  HYPERPARAMETERS

In the experiments with projecting to 3D-space we trained model for 100 epochs using Adam optimizer. We initially trained autoencoder for 10 epochs with only the reconstruction loss and learning rate 1e-4, then continued with RTD. Epochs 11-30 were trained with learning rate 1e-2, epochs 31-50 with learning rate 1e-3 and for epochs all after learning rate 1e-4 was used. Batch size was 80.

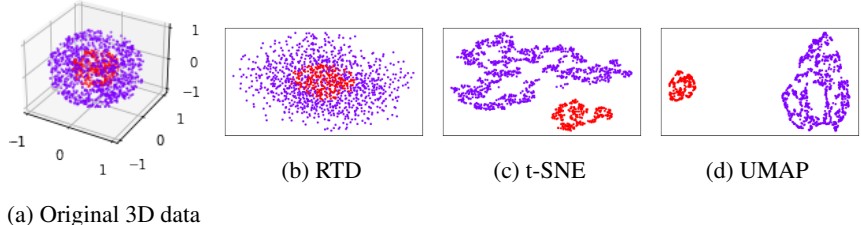

(a) Original 3D data     (b) RTD     (c) t-SNE     (d) UMAP

Figure 9: "Nested spheres" dataset.

Table 6: Quality of data manifold global structure preservation on synthetic data.

| Dataset | Method | Quality measure | | | | |
|---------|--------|-------|---------|---------|-------|------|
| | | L. C. | W. D. $H_0$ | W.D. $H_1$ | T. A. | RTD |
| Circle | t-SNE | 0.986 | 1.073 | 0.079 | 0.95 | 0.59 |
| | UMAP | 0.808 | 1.823 | 0.712 | 0.81 | 1.46 |
| | AE | 0.630 | 1.179 | 0.744 | 0.81 | 1.02 |
| | PaCMAP | 0.747 | 2.263 | N.A. | 0.81 | 1.61 |
| | Ivis | **0.990** | 0.182 | N.A. | **0.96** | 0.18 |
| | PHATE | 0.891 | 0.871 | N.A. | 0.88 | 1.04 |
| | TopoAE | 0.978 | 0.220 | 0.080 | 0.95 | 0.19 |
| | RTD-AE | 0.984 | **0.105** | **0.070** | **0.96** | **0.07** |
| 2 Clusters | t-SNE | 0.633 | 9.122 | 1.171 | 0.72 | 5.18 |
| | UMAP | 0.542 | 9.003 | 0.914 | 0.84 | 6.27 |
| | AE | 0.925 | 1.654 | 0.807 | **0.94** | 2.03 |
| | PaCMAP | 0.269 | 10.41 | N.A. | 0.64 | 5.71 |
| | Ivis | 0.423 | 7.400 | N.A. | 0.76 | 5.58 |
| | PHATE | 0.281 | 7.356 | N.A. | 0.66 | 4.90 |
| | TopoAE | 0.719 | 7.692 | 0.883 | 0.87 | 3.59 |
| | RTD-AE | **0.999** | **0.313** | **0.313** | **0.96** | **0.32** |
| 3 Clusters | t-SNE | 0.751 | 4.111 | 0.370 | 0.81 | 0.91 |
| | UMAP | 0.615 | 2.671 | 0.280 | 0.78 | 0.83 |
| | AE | 0.907 | 1.013 | 0.054 | 0.93 | 0.59 |
| | PaCMAP | 0.778 | 2.620 | N.A. | 0.89 | 0.92 |
| | Ivis | 0.918 | 2.511 | N.A. | 0.82 | 1.30 |
| | PHATE | 0.651 | 1.538 | N.A. | 0.72 | 0.99 |
| | TopoAE | 0.997 | 0.586 | 0.054 | 0.81 | 0.13 |
| | RTD-AE | **0.999** | **0.307** | **0.028** | **0.99** | **0.11** |
| Random | t-SNE | 0.981 | 4.182 | 1.938 | 0.95 | 1.54 |
| | UMAP | 0.950 | 0.979 | 0.622 | 0.91 | 0.55 |
| | AE | 0.700 | 9.976 | 2.343 | 0.75 | 1.32 |
| | PaCMAP | 0.982 | 5.398 | N.A. | 0.95 | 2.08 |
| | Ivis | 0.648 | 11.49 | N.A. | 0.75 | 2.17 |
| | PHATE | 0.945 | 6.703 | N.A. | 0.92 | 2.12 |
| | TopoAE | 0.854 | 3.288 | 1.367 | 0.84 | 0.91 |
| | RTD-AE | **0.996** | **0.148** | **0.389** | **0.98** | **0.17** |

For 2D and high-dimensional projections, we used fully-connected autoencoders with hyperparameters specified in the Table 7. The autoencoder was initially trained only with reconstruction loss for some number of epochs, and then the RTD loss kicked in. The learning rate stayed the same for an entire duration of training.

For experiments we used NVIDIA TITAN RTX.

## I  RTD OPTIMIZATION SPEED-UPS

For all computations of RTD-barcodes in this work we used modified version of Ripser++ software (Zhang et al., 2020). Modification that we made was intended at decreasing computational time via exploration of the structure of graph $\hat{\mathcal{G}}^{w,\tilde{w}}$ (see Section 3.2). The idea behind it is to reduce the size of filtered complex by excluding from it the simplices that do not affect the persistence homology.

Here we consider only simplices of dimension at least 1.

We exclude all simplices spanned by vertices from the first half of the vertex set of the graph $\hat{\mathcal{G}}^{w,\tilde{w}}$. Those are the vertices corresponding to the upper-left quadrant of the graph's edge weights matrix

Table 7: Hyperparameters description.

| Dataset name | Batch size | LR | Hidden dim | # layers | Epochs | RTD epoch |
|---|---|---|---|---|---|---|
| Circle | 80 | $10^{-3}$ | 16 | 3 | 100 | 20 |
| Random | 80 | $10^{-3}$ | 16 | 3 | 100 | 20 |
| 2 Clusters | 80 | $10^{-3}$ | 16 | 3 | 100 | 20 |
| 3 Clusters | 80 | $10^{-3}$ | 16 | 3 | 100 | 20 |
| Mammoth | 256 | $10^{-3}$ | 32 | 3 | 100 | 5 |
| MNIST | 256 | $10^{-4}$ | 512 | 3 | 250 | 60 |
| F-MNIST | 256 | $10^{-4}$ | 512 | 3 | 250 | 60 |
| COIL-20 | 256 | $10^{-4}$ | 512 | 3 | 250 | 60 |
| scRNA mice | 256 | $10^{-3}$ | 768 | 3 | 250 | 60 |
| scRNA melanoma | 256 | $10^{-3}$ | 768 | 3 | 250 | 60 |

$m$ from section B. All of them have diameters equal to zero. And if any such simplex spawn a topological feature, it is immediately killed by another such simplex.

As before, let $N$ be the number of vertices in point clouds. Then $\hat{\mathcal{G}}^{w,\tilde{w}}$ has $2N$ vertices and our modification eliminates $\binom{N}{d}$ out of $\binom{2N}{d}$ simplices of dimension $d-1$.

In particular, this eliminates around $1/8$ of rows and $1/4$ of columns (around $1/3$ cells in total) from the boundary matrix used for the computation of persistence pairs of dimension 1. On average, comparing to the standard Ripser++ computation, this gives $\approx 45\%$ less time for the computation of persistence intervals of dimension 1.

Next, we describe some techniques that can improve convergence when RTD is to be minimized without an autoencoder (F).

Usually we perform (sub)gradient descent to minimize $\text{RTD}(X, \tilde{X})$ between "movable" cloud $X$ and given constant $\tilde{X}$.

**Gradient smoothing.** Subgradients computed at each step of this procedure associate each homological class with at most 4 points from $X$, while topological structures often include much more. Moreover, adjustments w.r.t. them may be inconsistent for nearby points.

To overcome this, we "smooth" gradients by passing to each point averaged gradients of all its neighbours. Let $\nabla_i^{(k)}$ be the gradient value for $X_i$ at step $k$ and $U(X_i^{(k)})$ be some neighbourhood of $X_i^{(k)}$. Then the formula for each step of the gradient descent is

$$X_i^{(k)} = X_i^{(k-1)} - \lambda_k \left( \beta \nabla_i^{(k)} + (1-\beta) \frac{1}{\#\{X_j^{(k)} \in U(X_i^{(k)})\}} \sum_{X_j^{(k)} \in U(X_i^{(k)})} \nabla_j^{(k)} \right)$$

Here $\beta \in [0;1]$ is some parameter.

**Minimum bypassing.** Suppose we want to shorten an edge $m_{i+N,j+N}$ from bottom-right quadrant of matrix $m$ (i.e. $\frac{\partial \, \text{RTD}(X,\tilde{X})}{\partial m_{i+N,j+N}} < 0$). It may occur that $w_{i,j} > \tilde{w}_{i,j}$, so

$$\frac{\partial m_{i+N,j+N}}{\partial X_i} = \frac{X_i - X_j}{||X_i - X_j||_2} \mathbb{I}\{w_{i,j} < \tilde{w}_{i,j}\} = 0$$

and gradient descent will stuck here (since $\tilde{X}$ is constant). Thus there may appear a certain threshold below which $\text{RTD}(X, \tilde{X})$ can't be minimized in this case. But it *can* be further minimized if we move points $X_i$ and $X_j$ close enough to each other so $w_{i,j} < \tilde{w}_{i,j}$.

Table 8: RTD optimization speed-ups

| Optimization trics | RTD | Relative value (%) |
|---|---|---|
| None | 3.09059 | 100.00% |
| Gradient Smoothing | 2.68406 | 86.73% |
| Minimum Bypassing | 1.64366 | 53.07% |
| Both | 1.20738 | 38.84% |

To do it, if $\frac{\partial \text{ RTD}(X,\tilde{X})}{\partial m_{i+N,j+N}} < 0$ , we compute $\frac{\partial m_{i,j}}{\partial X_i}$ without indicator, i.e. as $\frac{X_i - X_j}{||X_i - X_j||_2}$. This will assure $w_{i,j}$ is decreasing and at certain point will became lower than $\tilde{w}_{i,j}$.

If $\frac{\partial \text{ RTD}(X,\tilde{X})}{\partial m_{i+N,j+N}} \geq 0$ we don't change anything, because the discussing effect appears only if we minimize a minimum of a function and a constant.

We performed an experiment to transform a cloud in the shape of the infinity sign by minimizing the RTD between this cloud and a ring-shaped cloud. Both clouds had 100 points and we did not use batch optimisation. We performed 100 iterations of gradient descend to minimize RTD in each of the following four setups: using none, each or both of Gradient Smoothing and Minimum Bypassing tricks. For each setup we also searched for the best learning rate. The Table 8 shows the results after 100 iterations.

## J    COMPARISON WITH TOPOAE LOSS

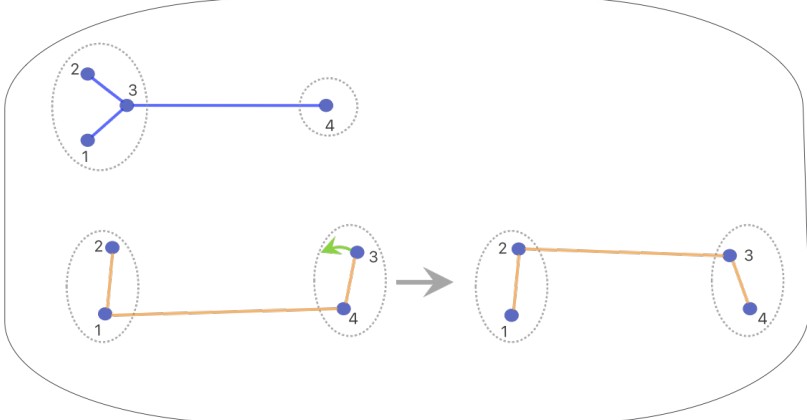

Figure 10: Discontinuity of the TopoAE loss. The point cloud $\tilde{X}$ consists of two clusters $\{1, 2, 3\}$ and $\{4\}$ (top). The point cloud $X$ (bottom left) consists of two clusters $\{1, 2\}$ and $\{3, 4\}$. (bottom left). The distances within each cluster are of order $10^{-1}$ and the distances between the clusters equal to $10^3 \pm 10^{-1}$. The TopoAE loss is discontinuous because under a small perturbation of points, the minimal spanning tree $\Gamma$ may change. When the point 3 moves slightly as indicated, then the minimal spanning tree $\Gamma$, coloured by yellow, changes and the term $(w_{14} - \tilde{w}_{14})^2 \sim 10^{-2}$ in TopoAE loss is replaced by $(w_{23} - \tilde{w}_{23})^2 \sim 10^6$.

The following simple example on Figure 10 shows that the TopoAE loss can be discontinuous in a rather standard situation. The TopoAE loss (Moor et al., 2020) is constructed by calculating first the two minimal spanning trees $\Gamma$, $\tilde{\Gamma}$ for each of the graphs $\mathcal{G}^w$, $\mathcal{G}^{\tilde{w}}$, whose weights are the distances within two point clouds $X$ and $\tilde{X}$. Then the TopoAE loss is the sum of two terms $L^{\text{TopoAE}} = l + \tilde{l}$. One term is the sum over the set of edges of $\Gamma$: $l = \frac{1}{2} \sum_{ij \in \text{Edges}(\Gamma)} (w_{ij} - \tilde{w}_{ij})^2$, and the other is the analogous sum over the edges of $\tilde{\Gamma}$: $\tilde{l} = \frac{1}{2} \sum_{ij \in \text{Edges}(\tilde{\Gamma})} (w_{ij} - \tilde{w}_{ij})^2$. Under a small perturbation of points, the minimal spanning tree $\Gamma$ may change, e.g. with a change of pair of the closest points

from two clusters. But then the corresponding weights $\tilde{w}$ change in general discontinuosly. The point cloud $\tilde{X}$ on Figure 10 consists of two clusters $\{1, 2, 3\}$ and $\{4\}$. The point cloud $X$ consists of two clusters $\{1, 2\}$ and $\{3, 4\}$. We set the distances within each cluster to be of order $10^{-1}$ and the distance between the clusters equal to $10^3 \pm 10^{-1}$. When the point 3 moves in $X$ slightly as indicated, then the minimal spanning tree $\Gamma$, coloured by yellow, changes and the term $(w_{14} - \tilde{w}_{14})^2 \sim 10^{-2}$ in $l$ is replaced by $(w_{23} - \tilde{w}_{23})^2 \sim 10^6$.

**Proposition 5.** *The RTD loss is continuous. The $RTD_k(X, \tilde{X})$ depends continuously on $(X, \tilde{X})$.*

The proof follows from the stability of the barcode of the filtered complex $\text{VR}_\alpha(\hat{\mathcal{G}}^{w, \tilde{w}})$ with respect to the bottleneck distance under perturbation of the edge weights, see Appendix K.

# K  STABILITY OF *R-Cross-Barcode* AND *RTD*

**Proposition 6.** *For any perturbations $X'$ of a point cloud $X$ and $\tilde{X}'$ of a point cloud $\tilde{X}$,*

$$d_B(\textit{R-Cross-Barcode}_k(X, \tilde{X}), \textit{R-Cross-Barcode}_k(X', \tilde{X}'))$$
$$\leq 2 \max(\max_i \|X_i' - X_i\|, \max_j \|\tilde{X}_j' - \tilde{X}_j\|) \quad (5)$$

*where $d_B$ denotes the bottleneck distance.*

*Proof.* By construction, the R-Cross-Barcode$_k(X, \tilde{X})$ is the $k-$th persistence barcode of the weighted graph $\hat{\mathcal{G}}^{w, \tilde{w}}$ with the weights $w_{ij} = \|X_i - X_j\|$ and $\min(w_{ij}, \tilde{w}_{ij})$, where $\tilde{w}_{ij} = \|\tilde{X}_i - \tilde{X}_j\|$. If $\max_i \|X_i' - X_i\| = \varepsilon$, then $|w_{ij}' - w_{ij}| \leq 2\varepsilon$ for $w_{ij}' = \|X_i' - X_j'\|$. Similarly, $|\tilde{w}_{ij}' - \tilde{w}_{ij}| \leq 2\tilde{\varepsilon}$, where $\tilde{\varepsilon} = \max_j \|\tilde{X}_j' - \tilde{X}_j\|$. It follows that $|\min(w_{ij}', \tilde{w}_{ij}') - \min(w_{ij}, \tilde{w}_{ij})| \leq 2 \max(\varepsilon, \tilde{\varepsilon})$. Hence the filtration of each simplex in $VR_\alpha(\hat{\mathcal{G}}^{w, \tilde{w}})$ changes at most by $2 \max(\varepsilon, \tilde{\varepsilon})$ under the perturbations. Next, it follows from e.g. the description of metamorphoses of canonical forms in (Barannikov, 1994) that the birth or the death of each segment in the $k-$th barcode of $\hat{\mathcal{G}}^{w, \tilde{w}}$ changes under such perturbations at most by $2 \max(\varepsilon, \tilde{\varepsilon})$. □

The above arguments give also the proof for the following stability result.

**Proposition 7.** *For any quadruple of edge weights sets $w_{ij}$, $\tilde{w}_{ij}$, $w_{ij}'$, $\tilde{w}_{ij}'$ on $\mathcal{G}$:*

$$d_B(\textit{R-Cross-Barcode}_k(w, \tilde{w}), \textit{R-Cross-Barcode}_k(w', \tilde{w}'))$$
$$\leq \max(\max_{ij}|w_{ij}' - w_{ij}|, \max_{ij}|\tilde{w}_{ij}' - \tilde{w}_{ij}|) \quad (6)$$

*where $d_B$ denotes the bottleneck distance and R-Cross-Barcode$_k(w, \tilde{w})$ denotes the persistence barcode for the weighted graph $\hat{\mathcal{G}}^{w, \tilde{w}}$.*

**Proposition 8.** *For any pair of edge weights sets $w_{ij}$, $\tilde{w}_{ij}$:*

$$\|\textit{R-Cross-Barcode}_k(w, \tilde{w})\|_B \leq \max_{ij}|w_{ij} - \tilde{w}_{ij}| \quad (7)$$

*where $\|\ \|_B$ denotes the bottleneck norm.*

*Proof.* Substitute $w' = \tilde{w}' = \tilde{w}$ into (6). □

Notice that (7) is analogous to (Barannikov et al., 2021, Proposition 1).

Given a pair of metrics $u, u'$ on a measure space $(\mathcal{X}, \mu)$, an analogue of Gromov-Wasserstein distance between $u$ and $u'$ is

$$GW(u, u') = \inf_{e, e': \mathcal{X} \hookrightarrow Z} \int_{\mathcal{X}} \rho_Z(e(x), e'(x)) \, d\mu \quad (8)$$

where $e : \mathcal{X} \hookrightarrow Z$, $e' : \mathcal{X} \hookrightarrow Z$ are embeddings to various metric spaces $(Z, \rho_Z)$ that are isometric with respect to $u, u'$.

**Proposition 9.** *Given a triple of metrics $u, u', \tilde{u}$ on a measure space $(\mathcal{X}, \mu)$, the expectation for the bottleneck distance between the R-Cross-Barcode$_k(w, \tilde{w})$ and the R-Cross-Barcode$_k(w', \tilde{w})$, comparing the pairs of weighted graphs associated with a sample $X = \{x_1, \ldots, x_n\}$, $x_i \in \mathcal{X}$, with the edge weights $w_{ij} = u(x_i, x_j)$, $w'_{ij} = u'(x_i, x_j)$, $\tilde{w}_{ij} = \tilde{u}(x_i, x_j)$, is upper bounded by the Gromov-Wasserstein distance between $u$ and $u'$:*

$$\int_{\mathcal{X} \times \ldots \times \mathcal{X}} d_B(\text{R-Cross-Barcode}_k(w, \tilde{w}), \text{R-Cross-Barcode}_k(w', \tilde{w})) \, d\mu^{\otimes n} \leq n \, GW(u, u') \quad (9)$$

*Proof.* It follows from the R-Cross-Barcode stability (6) that

$$\int_{\mathcal{X} \times \ldots \times \mathcal{X}} d_B(\text{R-Cross-Barcode}_k(w, \tilde{w}), \text{R-Cross-Barcode}_k(w', \tilde{w})) \, d\mu^{\otimes n} \leq$$

$$\leq \int_{\mathcal{X} \times \ldots \times \mathcal{X}} \max_{ij} |w_{ij} - w'_{ij}| \, d\mu^{\otimes n}.$$

For any pair of isometric embeddings $e : \mathcal{X} \hookrightarrow Z$, $e' : \mathcal{X} \hookrightarrow Z$:

$$|w_{ij} - w'_{ij}| = |\rho_Z(e(x_i), e(x_j)) - \rho_Z(e'(x_i), e'(x_j))| \leq$$

$$\leq \rho_Z(e(x_i), e'(x_i)) + \rho_Z(e(x_j), e'(x_j)) \leq \sum_{i=1}^{n} \rho_Z(e(x_i), e'(x_i))$$

by the triangle inequality for $\rho_Z$. Therefore

$$\int_{\mathcal{X} \times \ldots \times \mathcal{X}} d_B(\text{R-Cross-Barcode}_k(w, \tilde{w}), \text{R-Cross-Barcode}_k(w', \tilde{w})) \, d\mu^{\otimes n} \leq$$

$$\leq \int_{\mathcal{X} \times \ldots \times \mathcal{X}} \sum_{i=1}^{n} \rho_Z(e(x_i), e'(x_i)) \, d\mu^{\otimes n} = n \int_{\mathcal{X}} \rho_Z(e(x), e'(x)) \, d\mu$$

$\square$

**Proposition 10.** *The expectation for the bottleneck norm of R-Cross-Barcode$_k(w, \tilde{w})$ for two weighted graphs with edge weights $w_{ij} = u(x_i, x_j)$, $\tilde{w}_{ij} = \tilde{u}(x_i, x_j)$, where $u, \tilde{u}$ is a pair of metrics on a measure space $(\mathcal{X}, \mu)$, and $X = \{x_1, \ldots, x_n\}$, $x_i \in \mathcal{X}$ is a sample from $(\mathcal{X}, \mu)$, is upper bounded by Gromov-Wasserstein distance between $u$ and $\tilde{u}$:*

$$\int_{\mathcal{X} \times \ldots \times \mathcal{X}} \|\text{R-Cross-Barcode}_k(w, \tilde{w})\|_B \, d\mu^{\otimes n} \leq n \, GW(u, \tilde{u}) \quad (10)$$

*Proof.* Substitute $u' = \tilde{u}$, $w' = \tilde{w}$ into (9) $\square$

## L  DATASETS

The exact size, nature and dimension of the datasets are presented in Table 9. The errors for the synthetic data are not reported as they are zero due to the small sizes of the datasets.

### L.1  SYNTHETIC DATA

The "Random" dataset consists of 500 points randomly distributed on a 2-dimensional unit square. The choice for this dataset was inspired by Coenen & Pearce (2019a) and the ability of UMAP to find clusters in noise.

The "Circle" dataset is represented by 100 points randomly distributed on a 2D circle. This dataset has a simple non-trivial topology.

The "2 Clusters" dataset consists of 200 points, half of which goes to a dense Gaussian cluster, and the other half goes to sparse Gaussian cluster with the same mean. It is used to test the methods abilities to preserve cluster density.

The "3 Clusters" dataset consists of 3 Gaussian clusters each having 100 points. Two clusters are located much closer to each other than the remaining one. We propose it to test the preservation of the global structure, i.e. the distances between clusters.

Table 9: Datasets description.

| Dataset name | Total size | Nature | Dimension |
|---|---|---|---|
| Circle | $1 \times 10^2$ | Synthetic | 2 |
| Random | $5 \times 10^2$ | Synthetic | 2 |
| 2 Clusters | $2 \times 10^2$ | Synthetic | 2 |
| 3 Clusters | $3 \times 10^2$ | Synthetic | 2 |
| Mammoth | $50 \times 10^3$ | Real | 3 |
| F-MNIST | $70 \times 10^3$ | Real | 784 |
| COIL-20 | 1440 | Real | 16384 |
| scRNA mice | 1402 | Real | 25392 |
| scRNA melanoma | 4645 | Real | 23686 |

## L.2 REAL-WORLD DATASETS

Both MNIST and F-MNIST are typical datasets, consisting of 60000 28×28 pixel pictures of 10 different numbers and 10 types of clothes accordingly. COIL-20 is a dataset of pictures of 20 objects taken from 72 different angles spanning 360 degrees. scRNA mice dataset has 1402 single nuclei extracted from hippocampal anatomical sub-regions (DG, CA1, CA2, and CA3), and scRNA melanoma dataset monitors expression of 4645 cells isolated from 19 metastatic melanoma patients (Szubert et al., 2019).

Datasets licences:

- Mammoth (Coenen & Pearce, 2019b), CC Zero License. Mammuthus primigenius (blumbach), The Smithsonian Institute, *https://3d.si.edu/object/3d/mammuthus-primigenius-blumbach:341c96cd-f967-4540-8ed1-d3fc56d31f12*
- MNIST (LeCun et al., 1998), MIT License.
- Fashion-MNIST (Xiao et al., 2017), MIT License.
- COIL-20 (Nene et al., 1996).
- scRNA mice (Yuan et al., 2017).
- scRNA melanoma (Tirosh et al., 2016).

## M MORE DETAILS ON EXPERIMENTS WITH "SPHERES" AND "TORUS"

We performed experiments on dimensionality reduction to 3D space to evaluate preservation of 3-dimensional structures in data by our method. Experimental setup was outlined in Section 5.

For this task we have used two synthetic datasets.

The "Spheres" dataset consists of 17,250 points randomly distributed on surface of eleven 100-spheres in 101-dimensional space. Any two of those do not intersect and one of the spheres contains all other inside. Similar to "Circle" dataset (Section 5.3) UMAP splits bigger sphere (light grey) into 10 parts and wraps each small sphere into one of them. PacMAP performs similar but it also splits a part of bigger sphere into separate sphere. PCA and Ivis preserve the shape of inner spheres only and turn all structure 'inside out'. Both t-SNE and regular AE projects all points onto one sphere without clear separation between clouds. The addition of a topological loss, both in TopoAE and in our RTD-AE, preserves the global structure of inlaid clusters. However, TopoAE flattens inner clusters into disks, while RTD-AE makes them into (hollow) spheres.

The "Torus" dataset consists of 5,000 points randomly distributed on surface of a 2-torus ($T^2$) immersed into 100-dimensional space. Due to such nature of this dataset, PCA and MDS methods

Table 10: Quality of data manifold global structure preservation at projection Torus dataset from 100D into 3D space and Spheres dataset from 101D to 2D.

| Dataset | Method | Quality measure | | | | |
|---|---|---|---|---|---|---|
| | | L. C. | W. D. $H_0$ | W. D. $H_1$ | T. A. | RTD |
| Torus 2D | t-SNE | 0.989 | $1.021 \pm 0.07$ | $0.594 \pm 0.05$ | $0.896 \pm 0.01$ | $1.533 \pm 0.09$ |
| | UMAP | 0.955 | $2.052 \pm 0.15$ | $0.931 \pm 0.07$ | $0.931 \pm 0.07$ | $3.250 \pm 0.17$ |
| | PaCMAP | 0.987 | $1.410 \pm 0.12$ | $0.833 \pm 0.08$ | $0.883 \pm 0.01$ | $2.114 \pm 0.08$ |
| | PHATE | 0.873 | $2.967 \pm 0.33$ | $1.143 \pm 0.09$ | $0.646 \pm 0.02$ | $4.061 \pm 0.20$ |
| | PCA | **1.0** | $\mathbf{0.871 \pm 0.24}$ | $\mathbf{0.014 \pm 0.00}$ | $\mathbf{0.999 \pm 0.00}$ | $\mathbf{0.000 \pm 0.00}$ |
| | MDS | **1.0** | $\mathbf{0.880 \pm 0.24}$ | $\underline{0.022} \pm 0.00$ | $\mathbf{0.999 \pm 0.00}$ | $\mathbf{0.000 \pm 0.00}$ |
| | Ivis | 0.844 | $2.606 \pm 0.27$ | $1.086 \pm 0.11$ | $0.580 \pm 0.02$ | $4.073 \pm 0.17$ |
| | AE | 0.880 | $2.023 \pm 0.30$ | $1.073 \pm 0.08$ | $0.662 \pm 0.02$ | $3.433 \pm 0.12$ |
| | TopoAE | 0.920 | $2.616 \pm 0.34$ | $1.017 \pm 0.09$ | $0.696 \pm 0.02$ | $2.975 \pm 0.14$ |
| | RTD-AE | $\underline{0.992}$ | $\underline{0.907} \pm 0.08$ | $0.109 \pm 0.01$ | $\underline{0.902} \pm 0.01$ | $\underline{0.148} \pm 0.01$ |
| Spheres 2D | t-SNE | 0.018 | $49.77 \pm 1.40$ | $0.349 \pm 0.05$ | $0.166 \pm 0.01$ | $44.00 \pm 1.44$ |
| | UMAP | 0.020 | $\underline{47.55} \pm 1.33$ | $0.233 \pm 0.03$ | $0.191 \pm 0.01$ | $45.41 \pm 1.47$ |
| | PaCMAP | $\underline{0.342}$ | $\mathbf{46.57 \pm 1.68}$ | $\underline{0.208} \pm 0.02$ | $0.155 \pm 0.01$ | $45.56 \pm 1.46$ |
| | PHATE | 0.040 | $48.68 \pm 1.70$ | $\mathbf{0.188 \pm 0.03}$ | $0.201 \pm 0.01$ | $45.08 \pm 1.93$ |
| | PCA | 0.117 | $49.58 \pm 1.60$ | $0.447 \pm 0.05$ | $0.180 \pm 0.02$ | $43.01 \pm 1.36$ |
| | MDS | N.A. | N.A. | N.A. | N.A. | N.A. |
| | Ivis | 0.280 | $48.84 \pm 1.73$ | $0.342 \pm 0.05$ | $0.125 \pm 0.01$ | $44.21 \pm 1.36$ |
| | AE | 0.334 | $48.31 \pm 1.74$ | $0.320 \pm 0.04$ | $0.124 \pm 0.01$ | $43.74 \pm 1.60$ |
| | TopoAE | 0.264 | $49.94 \pm 1.52$ | $0.634 \pm 0.06$ | $\underline{0.245} \pm 0.02$ | $\underline{42.70} \pm 1.74$ |
| | RTD-AE | **0.611** | $48.20 \pm 1.72$ | $0.538 \pm 0.05$ | $\mathbf{0.343 \pm 0.01}$ | $\mathbf{41.22 \pm 1.70}$ |

Table 11: Quality of data manifold global structure preservation for projection of COIL-20 into 3D-space.

| Dataset | Method | Quality measure | | |
|---|---|---|---|---|
| | | L. C. | W. D. $H_0$ | T. A. |
| COIL-20 | t-SNE | 0.608 | $255 \pm 0.0$ | $0.706 \pm 0.01$ |
| | UMAP | 0.250 | $278 \pm 0.0$ | $0.574 \pm 0.012$ |
| | AE | 0.792 | $253 \pm 0.0$ | $0.803 \pm 0.009$ |
| | TopoAE | 0.677 | $236 \pm 0.0$ | $0.740 \pm 0.016$ |
| | RTD-AE | **0.811** | $\mathbf{233 \pm 0.0}$ | $\mathbf{0.814 \pm 0.014}$ |

perform on it very well. RTD-AE takes the third place with very similar quality, see Figure 11 and Table 10.

For "Spheres" dataset we have also performed experiments on dimensionality reduction to 2D space. Overall results are quite similar to those obtained for 3D case. The behavior of baselines remains essentially the same. The only interesting change is that RTD-AE now projects bigger sphere to a ring and puts the projections of smaller spheres into the ring's hollow center. RTD-AE outperforms other methods in terms of linear correlation and triplet accuracy.

All of the representations were generated with default parameters of baseline methods. Results are presented at Figures 4 ("Spheres" to 3D space) and 11 ("Spheres" to 2D space and "Torus" to 3D).

# N ABLATION STUDY

In this section we investigate the effect of adding RTD loss on the performance of the model. We add a hyperparameter $\lambda$ responsible for the scale of the RTD loss variable: $\mathcal{L}_{rec}(X, \tilde{X}) + \lambda \text{RTD}(X, Z)$. We run our experiments on two datasets: COIL-20 and Circle. The hyperparameter value ranged

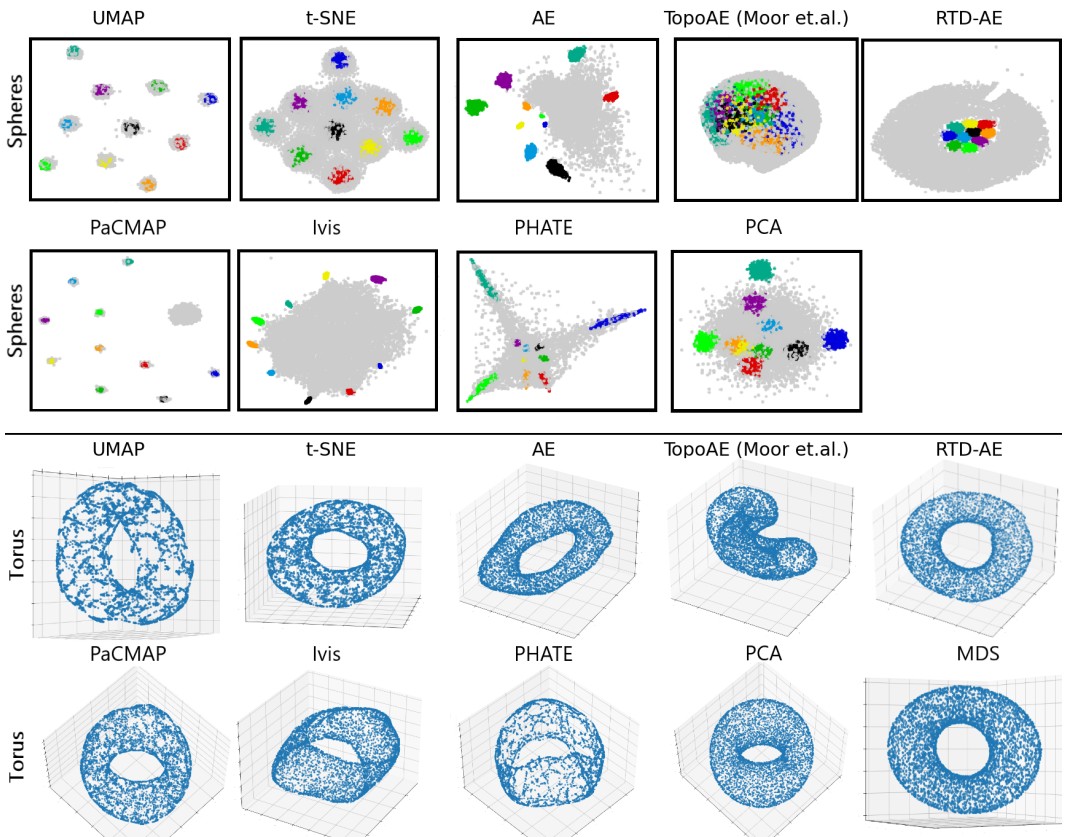

Figure 11: Results on dimensionality reduction of Spheres to 2D-space and Torus to 3D-space

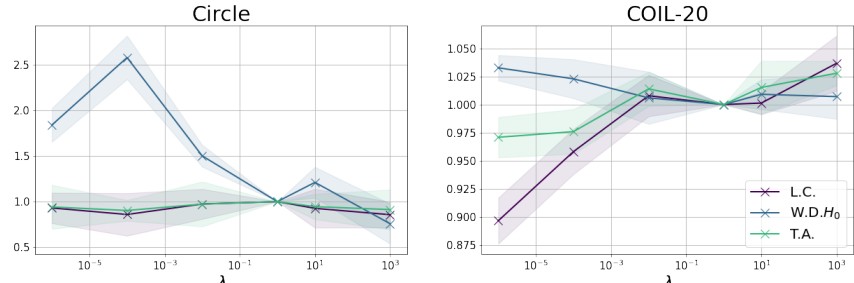

Figure 12: Results of ablation study. The plot depicts the value of the metrics relative to its value at $\lambda = 1.0$, which was used in all previous experiments. We clearly see that the addition of our RTD loss indeed increases linear correlation and triplet accuracy and at the same time decreases W.D.$H_0$. At the same time choosing $\lambda = 1.0$ seems reasonable to us as increasing its value further does not affect on the quality.

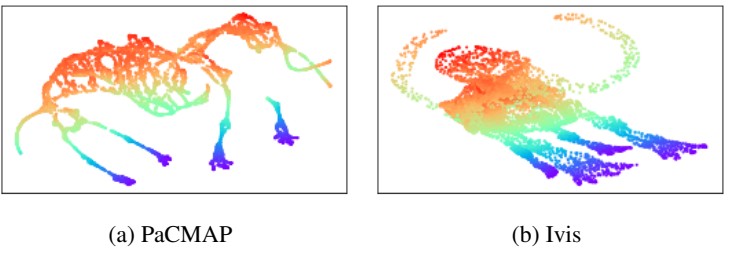

(a) PaCMAP          (b) Ivis

Figure 13: Additional dimensionality reduction methods applied to the "Mammoth" dataset

from $10^{-6}$ to $10^3$. For each value of $\lambda$ we run the procedure 8 times to get the confidence levels of our metrics. The results are depicted at Figure 12.

## O   MORE DIMENSIONALITY REDUCTION METHODS ON "MAMMOTH" DATASET

See Figure 13.

## P   R-CROSS-BARCODES

See Figure 14.

## Q   RECONSTRUCTION LOSS

See Table 12 for results.

## R   HYPERPARAMETERS SEARCH FOR SPHERES DATASET (INTO 2D)

For TopoAE we performed hyperparametrs search in accordance with the original paper Moor et al. (2020) and selected best combination according to $KL_{0.1}$-divergence.

For RTD-AE we searched for batch size in $[20; 250]$ and $\lambda$ in $[0.1; 10]$. Best combination was once again selected w.r.t. $KL_{0.1}$-divergence.

Results are presented in Table 13. For Wasserstein Distance and Triplet Accuracy difference between means is lesser than standard derivations, and due to this, we performed one-tailed Student's t-test to verify their relation. According to its results, we can reject the null hypothesis that the mean W.D.

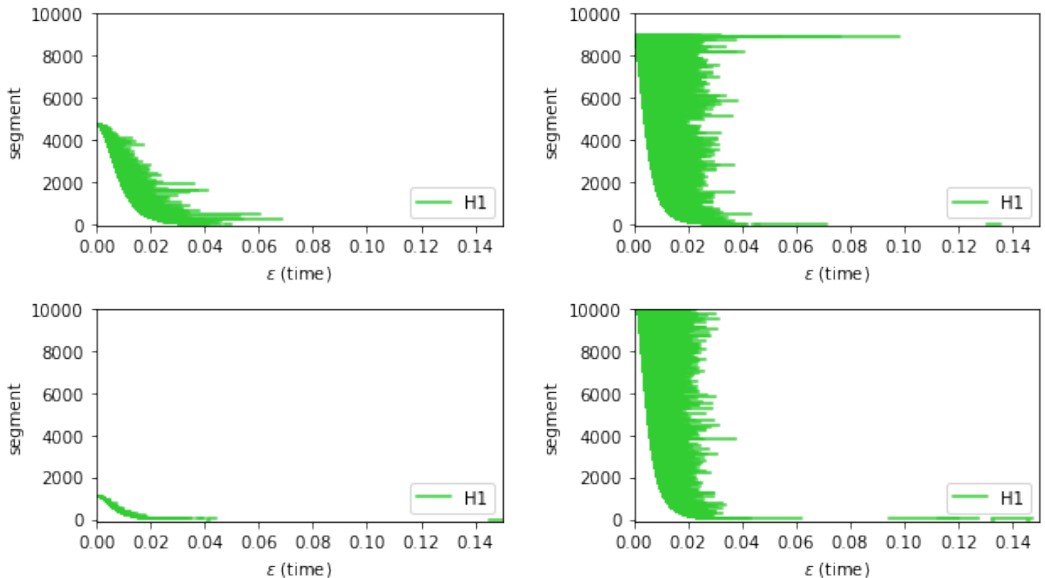

Figure 14: R-Cross-Barcodes between latent representations and original data points. Top: R-Cross-Barcode($Z_0, X$), R-Cross-Barcode($X, Z_0$). Bottom: R-Cross-Barcode($Z, X$), R-Cross-Barcode($X, Z$). $X$ - "Mammoth" dataset, $Z$ - latent representations from RTD-AE, $Z_0$ - latent representation from the untrained autoencoder. Intervals in R-Cross-Barcodes are smaller after training.

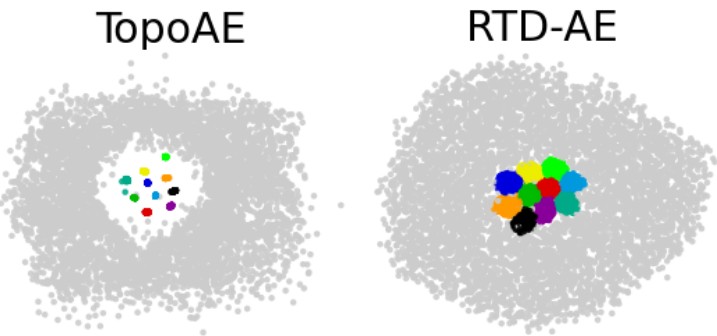

Figure 15: Dimensionality reduction of Spheres dataset to 2D-space after hyperparameter search.

$H_0$ for TopoAE is lower than the mean W.D. $H_0$ for RTD-AE at a significance level of 0.05. Same result confirming the better performance of RTD-AE was obtained for the triplet accuracy.

## S  ON IDENTITY OF INDISCERNIBLES FOR THE TOPOAE LOSS

We compare two point clouds $X, \tilde{X}$ from Figure 16. For these point clouds, RTD($X, \tilde{X}$) = 0.207, while the topological part of the TopoAE loss equals 0. The distinguishing topological feature between $X$ and $\tilde{X}$ is the cycle in $\tilde{X}$ which is born at $\alpha = 1$ and dies at $\alpha = \sqrt{2}$. R-Cross-Barcode($\tilde{X}$, X) depicts this difference.

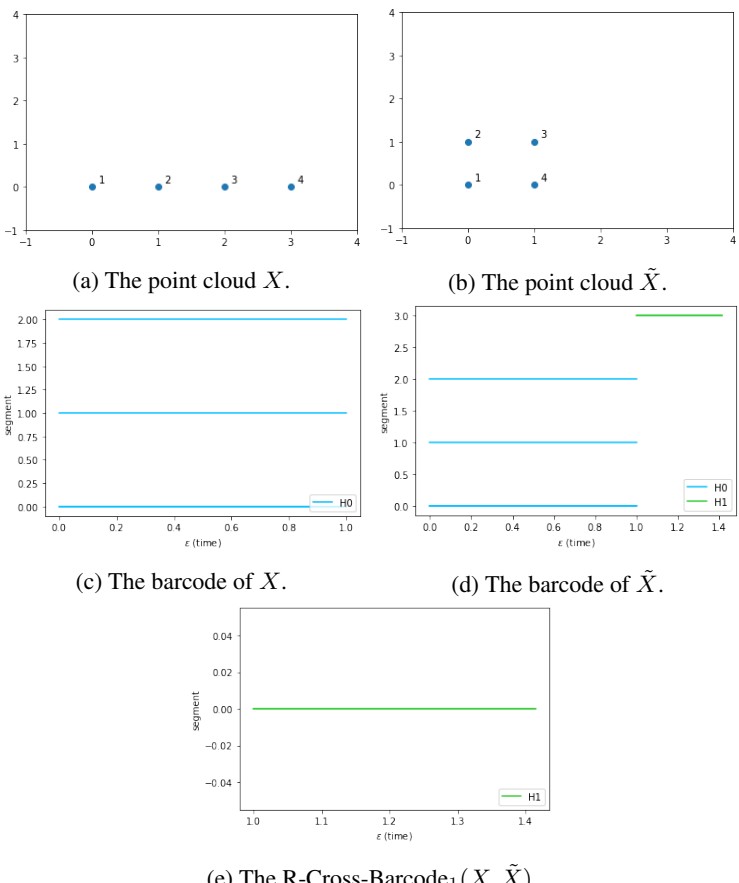

(a) The point cloud $X$.

(b) The point cloud $\tilde{X}$.

(c) The barcode of $X$.

(d) The barcode of $\tilde{X}$.

(e) The R-Cross-Barcode$_1(X, \tilde{X})$

Figure 16: Two point clouds $X, \tilde{X}$ for which the identity of indiscernibles property doesn't hold for the topological term in the TopoAE loss. The one-to-one correspondence between clouds is depicted by numbers. The minimal spanning trees $1 - 2 - 3 - 4$ have edges of identical length for both point clouds. For these point clouds, $\text{RTD}(X, \tilde{X}) = 0.207$, while the topological term of the TopoAE loss equals $0$. The topology of these point clouds is different, in particular they have different barcodes. The distinguishing topological feature between $X$ and $\tilde{X}$ is the cycle in $\tilde{X}$ which is born at $\alpha = 1$ and dies at $\alpha = \sqrt{2}$. The R-Cross-Barcode$_1(\tilde{X}, X)$ depicts this difference.

Table 12: Reconstruction loss for when projecting into 16 dimension latent space

| Dataset | Method | Reconstruction loss |
|---------|--------|---------------------|
| COIL-20 | AE | $1.89 \times 10^{-4}$ |
|         | TopoAE | $3.30 \times 10^{-4}$ |
|         | RTD-AE | $4.54 \times 10^{-4}$ |
| F-MNIST | AE | $2.73 \times 10^{-3}$ |
|         | TopoAE | $2.84 \times 10^{-3}$ |
|         | RTD-AE | $3.17 \times 10^{-3}$ |
| MNIST | AE | $3.78 \times 10^{-3}$ |
|       | TopoAE | $3.70 \times 10^{-3}$ |
|       | RTD-AE | $4.88 \times 10^{-3}$ |
| scRNA mice | AE | $1.31 \times 10^{-3}$ |
|            | TopoAE | $1.23 \times 10^{-3}$ |
|            | RTD-AE | $1.32 \times 10^{-3}$ |
| scRNA melanoma | AE | $1.16 \times 10^{-3}$ |
|                | TopoAE | $1.11 \times 10^{-3}$ |
|                | RTD-AE | $1.15 \times 10^{-3}$ |

Table 13: Hyperparameter search for Spheres (into 2D) dataset

| Dataset | Method | L.C. | W.D. $H_0$ | T.A. | RTD |
|---------|--------|------|------------|------|-----|
| Spheres 2D | TopoAE | 0.691 | $43.291 \pm 1.583$ | $0.3688 \pm 0.0165$ | $39.837 \pm 1.318$ |
|            | RTD-AE | **0.706** | **$42.133 \pm 1.683$** | **$0.3765 \pm 0.0124$** | **$37.286 \pm 1.393$** |

