# OpenReview forum: "Learning topology-preserving data representations"
_ICLR.cc/2023/Conference — ICLR 2023 poster_

### Official Review · Reviewer_LAR4 · 2022-10-17

**Confidence:** 4
**Correctness:** 4
**Technical Novelty And Significance:** 4
**Empirical Novelty And Significance:** 4
**Recommendation:** 8

**Clarity, Quality, Novelty And Reproducibility:**

The paper is well-written and clear. I find that key materials supporting the paper are present in the paper. Reproducing of the work is not an issue.

One minor issue is the double usage of symbol k, both as the dimensionality of simplices and as the dimensionality of the compressed data.

It could be interesting to know if RTD can be approximated or upper-bounded by a more easily differentiable loss function and how that can affect the output data of RTD-AE.

**Strength And Weaknesses:**

### Weaknesses

There is not much in terms of theoretical contribution. My impression is that it is mainly about making RTD work with differentiation, since RTD has been shown quite a useful divergence. In contrast, I acknowledge that it can be a very intensive engineering challenge.

The authors provided optimization tricks in Appendix I, including restricting the simplices of interest to just edges (k=1), gradient smoothing via averaging nearby subgradients, and bypassing plateaus when $w_{i,j} > \tilde{w}_{i,j}$. Given that topology-based optimization is typically very compute-intensive, often in cubic time, I was hoping to see some further analysis on how each of those tricks contributed to the learning of RTD-AE, in terms of changes in accuracy and/or speed. However, none was offered.

### Strength

Appendix J pointing out why the TopoAE loss is discontinuous is good to have. It would be good to also give an example why the TopoAE loss does not guarantee identity of indiscernibles.

Appendix M shows the effect of adding the RTD loss on linear correlation, triplet accuracy, and Wasserstein distance is. I find it helpful in understanding RTD.

The results show that RTD-AE outperform TopoAE in almost all settings.



**Summary Of The Paper:**

The paper proposes a new dimensionality reduction method, named RTD-AE, that attempts to preserve the topological structure of the data when they are compressed to a lower-dimensional space. This is done in a standard autoencoder fashion where the loss function consists of a reconstruction loss and a topology-preserving loss. The closest rival to RTD-AE is TopoAE (Moor et al, 2020) but it comes with a discontinuous topology-preserving loss that does not guarantee identity of indiscernibles (d(x,y) = 0 does not mean x = y). Instead, RTD-AE uses the Representation Topological Divergence (RTD) (Barannikov et al, 2022), which is continuous and guarantees identity of indiscernibles. However, RTD is not C1-smooth so the paper's main contribution is RTD differentiation using subgradients.

Experiments on synthetic datasets and practical datasets MNIST, F-MNIST, COIL-20, scRNA mice and scRNA melanoma, where the proposed method, named RTD-AE, is compared against recent methods like UMAP, t-SNE, a vanilla autoencoder (AE), TopoAE, Ivis, PacMAP and PHATE, show favouring results towards RTD-AE.

**Summary Of The Review:**

This is a good, well-rounded paper. My only concern is that for this conference I would expect the contribution to be more theoretical, although I appreciate the possibly large amount of efforts put in place to make RTD differentiation and RTD-AE work. However, the actual results mostly favouring RTD-AE over TopoAE and more distant methods is encouraging.

---

> ### Author Response · Authors · 2022-11-18
> **Response to the Review LAR4**
>
> Thank you for your time and positive feedback. We will improve the presentation according to the suggestions. Below we address specific concerns one by one.
>
> **Q1**: _There is not much in terms of theoretical contribution._
> **A**: In addition to the constructed approach to RTD differentiation that permits optimizing a data representation so that its multiscale k-dimensional topological features (clusters, 1-cycles, k-voids etc) become similar to that of a given data representation for any dimension k≥0, bypassing the standard scheme comparing two PH-diagrams, we described the following theoretical results.   We have shown the discontinuity of the TopoAE loss. We have proven the continuity of the RTD loss,  please see Appendix J.  We have added  the  theoretical result on the stability of the  R-Cross-Barcode$(X,\tilde{X}$), please see Appendix K.
>
>
>
>
> **Q2**: _Given that topology-based optimization is typically very compute-intensive, often in cubic time, I was hoping to see some further analysis on how each of those tricks contributed to the learning of RTD-AE, in terms of changes in accuracy and/or speed._
> **A** : Please see the speed gains description in Appendix I. We use the fact that the upper-left quadrant of the graph $\hat{\mathcal{G}}^{w, \tilde{w}}$ distance matrix is always 0, and eliminate the majority of simplices from the computation.  We get ~x2 speedup. Other tricks, namely the plateaus bypassing and gradient smoothing were not applied in experiments with autoencoders.
> These tricks are intended for the case when we directly minimize RTD between two clouds without autoencoder (setup of the Appendix F).
> We are adding more details about the speed gains in the case of optimization without AE. For example,  we performed an experiment transforming a cloud in shape of infinity sign  minimizing RTD between it and a ring-shaped cloud. Both clouds had 100 points and we did not use batch optimisation. We performed 100 iterations of gradient descend to minimize RTD in each of four setups (using none, each or both of Gradient Smoothing and Minimum Bypassing). For each instance we also searched for the best learning rate. Following table shows the results in terms of RTD after all 100 iterations.
> | Optimization tricks   | RTD  | Relative value (%) |
> |---|:---:|:---:|
> | None  |  3.09059  | 100.00%   |
> | Gradient Smoothing |  2.68406 | 86.73%  |
> | Minimum Bypassing |  1.64366 | 53.07%  |
> | Both |  1.20738 | 38.84%  |
>
> As one can see,  there is some synergy: combination of both tricks gives better relative improvement (in %) than the product of improvements from each trick.
>
>
>
>
> **Q3**: _Example why the TopoAE loss does not guarantee identity of indiscernibles._
> **A**: We have added in Appendix S the toy example when the TopoAE topological loss equals zero while the topology of two point clouds is distinct. The first point cloud $X$ consists of 4 points located consequently on a line at unit steps. The second point cloud $\tilde{X}$ consists of 4 points located in the vertices of a unit square in the clockwise order. For these point clouds, RTD=0.207, while the topological term of the TopoAE loss is zero.
> The distinguishing topological feature between the point clouds is the cycle in $\tilde{X}$ which is born at $\alpha=1$ and dies at $\alpha=\sqrt{2}$. The R-Cross-Barcode$_1(X,\tilde{X}$) depicts this difference, see Figure 16.
>
> **Q4**: _It could be interesting to know if RTD can be approximated or upper-bounded by a more easily differentiable loss function and how that can affect the output data of RTD-AE._
> **A**:  There exists indeed an approximation to RTD$_1$ which is sufficient for certain purposes, we are testing it in different scenarios, on some tasks it gives up to 10x speedup without much loss in quality, on other tasks the quality is lower, the details will appear in an upcoming work. Thank you for this thoughtful question.
>
>
> _Concluding remarks._ Please respond to our post to let us know if the clarifications above suitably address your concerns about our work. We are happy to address any remaining points during the discussion phase; if the responses above are sufficient, we kindly ask that you consider raising your score.

---

> > ### Comment · Reviewer_LAR4 · 2022-11-22
> > **Thank you.**
> >
> > I would like to thank the authors for the response. Answers to Q2..Q4 have addressed my feedback/concerns. Much appreciated.
> >
> > Regarding Q1, thanks for the answer but unfortunately it does not really make me think more highly of the contributions than before. All the contributions related to subgradients to make RTD differentiable (or approximately differentiable) in an efficient manner are not under-appreciated. However, to me they should be counted as engineering contributions rather than as theoretical contributions. The theoretical contributions I see are: (1) pointing out that TopoAE is dis-continuous, and (2) proof that RTD is continuous, both of which you described in appendix J in the first draft. The extension that you have introduced in appendix K in the new draft is welcome. It explains better why RTD is continuous. However, to me (2) is not hard to see just by following the definition of RTD in Barannikov et al's paper. Your 2-liner proof that RTD is continuous in the first draft seems to suggest that as well. In contrast, (1) is indeed new to me.
> >
> > This is why I wrote what I wrote in the review. But I must stress again that it does not mean that the engineering contributions are not appreciated.
> >
> > That said, I have changed the Technical Novelty and Significance rating from 2 to 3.

---

> > > ### Author Response · Authors · 2022-12-04
> > > **On the theoretical contribution (Q1)**
> > >
> > > Thank you for the feedback.
> > >
> > > You have pointed out that the theoretical result on stability mentioned in section 3.2 and described in Appendix K is welcomed, but still not enough from your point of view, - permit us to respectfully disagree. From our viewpoint, this theoretical result is the basic stability for the R-Cross-Barcode. It implies numerous immediate consequences. Here is a non-exhaustive list:
> > >
> > > _Proposition A._
> > > For any quadruple of edge weights sets $v _ {ij}$, ${\tilde{v}} _ {ij}$, $w _ {ij}$, ${\tilde{w}} _ {ij}$ :
> > > $d_B(\text{R-Cross-Barcode} _ k(w,\tilde{w}), \text{R-Cross-Barcode} _ k(v,\tilde{v})) \leq \max(\max _ {ij} \lvert v _ {ij}-w _ {ij}\rvert ,\max _ {ij}\lvert \tilde{v} _ {ij}-\tilde{w} _ {ij}\rvert)$.
> > >
> > > _Proposition B._
> > > For any  pair of edge weights sets ${w} _ {ij}$, $\tilde{w} _ {ij}$ :
> > >  $\lVert \text{R-Cross-Barcode} _ k(w,\tilde{w})\rVert _ B \leq \max _ {ij} \lvert w _ {ij}-\tilde{w} _ {ij}\rvert$.
> > >
> > > _Proposition C._
> > > The expectation for the bottleneck distance between the $\text{R-Cross-Barcode} _ k(w,\tilde{w})$ and the $\text{R-Cross-Barcode} _ k(w ' ,\tilde{w})$, comparing two pairs of weighted graphs
> > >     with the edge weights $w _ {ij}=w(x _ i,x _ j)$, $w ' _ {ij}=w ' (x _ i,x _ j)$, $\tilde{w} _ {ij}=\tilde{w}(x _ i ,x _ j)$, where $w,w ' ,\tilde{w}$ is a trple of metrics on a measure space $(\mathcal{X},\mu)$,
> > > and  $X=$&lcub;$x _ 1,\ldots, x _ n$&rcub; , $x _ i\in\mathcal{X}$ is a sample from $(\mathcal{X},\mu)$,  is upper bounded by Gromov-Wasserstein distance between $w$ and ${w} ' $:
> > >     $\int_{\mathcal{X}\times\ldots\times\mathcal{X}}d_B(\text{R-Cross-Barcode} _ k(w,\tilde{w}),\text{R-Cross-Barcode}_k(w ' ,\tilde{w}))  d\mu^{\otimes n }\leq n$    $GW(w,{w} ' )$.
> > >
> > > _Proposition D._
> > >  The expectation for the bottleneck norm of the $\text{R-Cross-Barcode} _ k(w,\tilde{w})$ for two weighted graphs with  edge weights  $w _ {ij}=w(x _ i,x _ j)$, $\tilde{w} _ {ij}=\tilde{w}(x _ i,x _ j )$, where $w ,\tilde{w}$ is a pair of metrics on a measure space $(\mathcal{X},\mu)$, and    $X  =$  &lcub;$x _ 1,\ldots, x _ n$&rcub;, $x _ i\in\mathcal{X}$ is a sample from $(\mathcal{X},\mu)$,
> > >      is upper bounded by Gromov-Wasserstein distance between $w$ and $\tilde{w}$:
> > > $    \int_{\mathcal{X}\times\ldots\times\mathcal{X}}\lVert \text{R-Cross-Barcode}_k(w,\tilde{w})\rVert_B  d\mu^{\otimes n }\leq n$&nbsp;$GW(w,\tilde{w})$.
> > >
> > > We are adding these results to Section 3 and their proofs to Appendix K.   B and D follow, respectfully, from  A and C via setting $v=\tilde{v}=\tilde{w}$ and $w ' =\tilde{w}$. The proof of A is mentioned in Appendix K.
> > > _The proof of C_ follows from A and the triangle inequality for the auxiliary metrics $\rho_Z$ from the definition of GW distance:
> > >
> > > For  a pair of metrics $w,{w} ' $ on a measure space $(\mathcal{X},\mu)$, the GW distance is
> > >  $GW(w,{w} ' )=\inf _ {e,{e} ' :\mathcal{X}\hookrightarrow Z}  \int _ {\mathcal{X}}\rho_Z(e(x),{e} ' (x)) d\mu$
> > >  where $e:\mathcal{X}\hookrightarrow Z$, ${e} ' :\mathcal{X}\hookrightarrow Z$ are  embeddings to various metric spaces $(Z,\rho _ Z)$ that are isometric with respect to $w$ and ${w} ' $. It follows from (A) that
> > > $\int_{\mathcal{X}\times\ldots\times\mathcal{X}}d _ B(\text{R-Cross-Barcode} _ k(w,\tilde{w}),\text{R-Cross-Barcode} _ k(w ' ,\tilde{w}))  d\mu^{\otimes n }\leq   \int _ {\mathcal{X}\times\ldots\times\mathcal{X}} \max _ {ij} \lvert w _ {ij}-{w} ' _ {ij}\rvert    d\mu^{\otimes n }.$
> > > For any pair of isometric embeddings $e:\mathcal{X}\hookrightarrow Z$, ${e} ' :\mathcal{X}\hookrightarrow Z$, by the triangle inequality for $\rho_Z$ :
> > > $\lvert w _ {ij}-{w} ' _ {ij}\rvert=\lvert\rho_Z(e(x _ i),{e}(x _ j)) -\rho _ Z(e ' (x _ i),{e} ' (x _ j))\rvert\leq \rho_Z(e(x _ i),{e} ' (x _ i)) +\rho _ Z(e(x _ j),{e} ' (x _ j)) \leq \sum _ {i=1} ^ n \rho_Z(e(x _ i),{e} ' (x _ i))$
> > > Hence $\int _ {\mathcal{X}\times\ldots\times\mathcal{X}}d _ B(\text{R-Cross-Barcode} _ k(w,\tilde{w}),\text{R-Cross-Barcode} _ k(w ' ,\tilde{w}))  d\mu^{\otimes n }\leq  \int _ {\mathcal{X}\times\ldots\times\mathcal{X}}\sum _ {i=1} ^ n \rho _ Z(e(x _ i),{e} ' (x _ i)) d\mu^{\otimes n }$
> > >  $= n \int_{\mathcal{X}} \rho_Z(e(x),{e}'(x)) d\mu$. $\square$
> > >
> > > Let us also point out the theoretical result describing the $\max-$variant of the R-Cross-Barcode, for details see Appendix G.
> > >
> > >
> > > _Concluding remarks._ Please respond to our post to let us know if the clarifications above suitably address your concerns about our work. We are happy to address any remaining points during the discussion phase; if the responses above are sufficient, we kindly ask that you consider raising your score.

---

> > > > ### Comment · Reviewer_LAR4 · 2022-12-07
> > > > **Thanks!**
> > > >
> > > > Thanks for the latest feedback.
> > > >
> > > > These are simple propositions and proofs.
> > > >
> > > > Still, they make the theoretical part of the paper stronger. Thus, assuming all these changes will be included in the final version:
> > > >
> > > > 1. I have increased the Technical Novelty And Significance rating from 3 to 4.
> > > > 2. I have increased the overall rating from 6 to 8.

---

### Official Review · Reviewer_XfmV · 2022-10-18

**Confidence:** 5
**Correctness:** 3
**Technical Novelty And Significance:** 3
**Empirical Novelty And Significance:** 3
**Recommendation:** 8

**Clarity, Quality, Novelty And Reproducibility:**

## Clarity

- I have some trouble understanding the turn of phrase "features are located in the same places". I think this refers to the sequences and their respective parameters, meaning that two features arise at roughly the same thresholds. If so, a brief explanation would be warranted.

- I would suggest turning the comparison with this "TopoAE" method into a separate paragraph/section instead of already providing details about it in the introduction.

- Since the RTD is being prominently used, Section 3.2 should be extended to provide additional details about the method. The appendix and cited reference are both useful, but the paper should also be understandable on its own without referring to external sources.

- When giving an example in Section 3.2, please use the aforementioned terminology for point clouds, i.e. $X$ and $\widetilde{X}$, instead of introducing a new one.

- The colours in Figure 2 are hard to distinguish for me. I appreciate such a graphical example very much, and would thus urge to update the edge colours a little bit so that the calculation of the cross barcode becomes more apparent.

- I would suggest rephrasing  the use of the term "nullity". As I read the manuscript, I understand the discussion after Proposition 1 to mean that the RTD loss guarantees a kind of metric "identity of indiscernibles", i.e. it is zero if and only if the two barcodes are equal. Maybe it would be easier to rephrase the text slightly here.

- In Section 4.1, $\hat{X}$ should be $\tilde{X}$, I think.

- The discussion of the matrix in Section 4.1 could be simplified. First, I would suggest using a different symbol for the matrix, potentially a bold font; this would make the matrix stand out a little bit from the existing text. Second, the indices seem to follow a different terminology than the one shown in Section 3.1. I prefer $ij$ instead of the $A_i A_j$ of Section 3.1, but this should be homogenised.

- Consider using a symbol to refer to the Wasserstein distance instead of using W.D.

## Quality

The paper is already of high quality. I particularly appreciate the detailed supplements, which serve to provide a further glimpse into the different aspects of the method. Thus, the proposed method has a strong theoretical foundation and clearly improves over the state of the art. However, when *evaluating* the proposed method, there are some issues that need to be rectified:

- Using RTD as its own quality measure strikes me as slightly tautological; I understand that the loss term optimises a very related quantity, and I would therefore suggest to also show additional quality metrics (even post-hoc ones) in the respective tables. For instance, one metric that would be highly relevant to discuss here would be the actual reconstruction loss of the autoencoder. Does the reconstruction quality suffer when using the new method? Moreover, what about metrics that are not directly related to topological quantities of the data, such as a distance correlation? At the moment, the table describing the experiments are slightly favouring topology-based methods. This should be avoided, and other metrics should be considered.

- When comparing to the other topology-based methods such as UMAP and TopoAE, and extension of Section 5.5 would be useful. I understand that the existing methods have some theoretical disadvantages (in particular the discontinuity of the TopoAE), but considering the computational complexity of the proposed approach, are there any trade-offs to existing methods to be discussed? This would only require a few additional lines; to save more space, some of the tables could be relegated to the appendix; personally, I am fine with having a selection of experiments in the main paper, with a more detailed description being provided in the supplements.

## Novelty

- For the delineation to existing research, the paper [*Optimizing persistent homology based functions*](http://proceedings.mlr.press/v139/carriere21a.html) should be more prominently discussed. To the best of my knowledge, this paper provides a general justification and explanation of how to differentiate and optimise topology-based functions. If so, please adjust the statement in the introduction. Similarly, a recent preprint by Wagner et al. on [*Improving Metric Dimensionality Reduction with Distributed Topology*](https://arxiv.org/abs/2106.07613) could be mentioned briefly as it also provides some hints regarding gradient calculations and optimisation in general.

## Reproducibility

The paper is very reproducible; I appreciated that the authors are taking the time to provide the source code of their method as well. Moreover, given the description of the algorithm, a skilled graduate student should be able to re-implement the method.

## Minor issues

- In the introduction: T-SNE --> t-SNE

- The t-SNE paper by 'Van der Maaten & Hinton' is cited twice in the first sentence of the related work section.

- Consider using `\operatorname` or a related LaTeX command for typesetting operators such as the image or the kernel; this improves readability of some equations.

- "Consider R-Cross-Barcode" --> "Consider the R-Cross-Barcode" (there are other places where a definite article could be added; I would suggest another pass over the paper)

- "having similar to $X$ topology" --> "having a topology similar to $X$"

- In certain places, the manuscript uses `\citet` or a plain `\cite` where a `\citep` would be more appropriate. For instance, in the paragraph "Comparison with TopoAE loss", the citation to the TopoAE method should be given in parentheses.

- Below Figure 3: "Let [...] denotes" --> "Let [...] denote"

- Please check the bibliography for consistent spelling and capitalisation. For instance, it should be "Carrière", not "Carriere", and "UMAP" instead of "umap". Moreover, the paper by Moon et al. seems to be cited twice; once as a preprint, the other time as a published version.



**Strength And Weaknesses:**

The paper makes a strong case for a new topology-based loss term. I appreciate the ingenious use of RTD, which seems to be perfectly apt for the task at hand (permitting the comparison of point clouds of the same size, among other things). Having a strong theoretical foundation with representation quality guarantees thus makes this a very strong contribution to the literature.

I have two minor issues with the manuscript in its current form, though:

1. Some details about RTD and its usage are missing; this should be rectified for a revision (please refer to the "Clarity" section below for more details).
2. The experimental setup needs to be strengthened by adding at least one or two "geometrical" metrics for the representation quality (please refer to the "Quality" section below for more details).

Apart from that, there are several minor issues, which I am confident can be addressed using some additional rewriting and rephrasing.


**Summary Of The Paper:**

This paper proposes a new topology-based loss term for dimensionality reduction. The loss term is based on the recently-introduced "Representation Topology Divergence" (RTD) and has beneficial theoretical as well as empirical properties. Next to comparing the loss term with existing losses arising in the context of topological machine learning, the paper presents a very detailed suite of experiments that showcase the empirical utility of the method.


**Summary Of The Review:**

This is a high-quality paper with a substantial contribution to the literature, namely a new topology-based loss term that satisfies advantageous theoretical and empirical properties. Apart from minor issues (see above), I am very happy to endorse this paper for publication. I am confident that the proposed changes can be performed within the revision cycle.

---

> ### Author Response · Authors · 2022-11-18
> **Response to the Review XfmV**
>
> Thank you for the thorough review and high evaluation of our work. We will improve the presentation according to the suggestions. Below we address specific concerns one by one.
>
> **Q1**: _The experimental setup needs to be strengthened by adding at least one or two "geometrical" metrics for the representation quality. What about metrics that are not directly related to topological quantities of the data, such as a distance correlation._
> **A**: Out of the four used metrics, we use two metrics precisely of this type: the linear correlation of pairwise distance (L.C.) and the triplet distance ranking accuracy (T.A.), please see, e.g. the first and the third columns in Tables 1-3.
>
>
> **Q2**: _Trouble understanding the turn of phrase "features are located in the same places"_
> **A**: It means that the topological features (clusters, 1-cycles, etc)  are localized at essentially  the same set of points, in the sense of the one-to-one correspondence,  inside clouds $X$ and $\tilde{X}$.  E.g. if a set of points $A_1,\ldots,A_n$ forms a separate cluster in the cloud $X$ at some threshold $\alpha$, then the corresponding points $\tilde{A}_1,\ldots,\tilde{A}_n$ form at the threshold $\alpha$ a separate cluster in $\tilde{X}$.   We are expanding a clarification on this.
>
>
>
> **Q3**: _Using RTD as a quality measure_. _Moreover, what about metrics  such as a distance correlation_.
> **A**: Besides RTD, we use 3 quality measures:  (1) linear correlation (L.C.) of pairwise distances, (2) Wasserstein distance (W.D.) between $H_0$ barcodes (3) triplet distance ranking accuracy (T.A.), please see Tables 1-6.  We note that RTD, as a quality measure, provides a more precise comparison of topology than the W.D. between $H_0$ persistence barcodes. First, RTD takes into account the localization of topological features, while W.D. does not. Second, W.D. is invariant to permutations of points, while we are interested in comparison between original data and latent representation where the natural one-to-one correspondence holds. We discuss this in Section 5.
>
> **Q4**: _For instance, one metric that would be highly relevant to discuss here would be the actual reconstruction loss of the autoencoder_.
> **A**: We have added the reconstruction loss to the paper, see Appendix P. The extra loss term in this experiment does not really change the final reconstruction loss which stays essentially zero.
>
> **Q5**: _Considering the computational complexity of the proposed approach, are there any trade-offs to existing methods to be discussed?_
> **A**: Computational experiments show that the proposed RTD-AE better
> preserves the global structure of the data manifold (as measured by linear correlation of pairwise distances, triplet distance
> ranking accuracy, Wasserstein distance between persistence barcodes), in particular, e.g. cluster densities, intercluster distances, or localization of topological features. We discuss this in Section 5.3, 6 and Appendix L.
>
> **Q6**: _For the delineation to existing research, the paper "Optimizing persistent homology based functions" should be more prominently discussed._
> **A**: Thank you for this remark. We have cited this paper on page 3 in the related work section, and  we are happily extending this.
>
> **Q7**: _A recent preprint by Wagner et al. on "Improving Metric Dimensionality Reduction with Distributed Topology" could be mentioned briefly._
> **A**: Thank you for attracting our attention to this paper, we are gladly adding it to the references.

---

> > ### Comment · Reviewer_XfmV · 2022-11-18
> > **Thanks**
> >
> > Thanks for addressing these issues. I'll take them into account during the upcoming discussion phase with my fellow reviewers.

---

### Official Review · Reviewer_f3NA · 2022-10-23

**Confidence:** 4
**Correctness:** 3
**Technical Novelty And Significance:** 4
**Empirical Novelty And Significance:** 3
**Recommendation:** 8

**Clarity, Quality, Novelty And Reproducibility:**

The paper is very well-written and easy to follow. I believe the idea is very interesting, novel and important.

**Strength And Weaknesses:**

**Strength**

The main idea of the paper as well as the empirical part are very interesting.

**Weaknesses and Questions**:

1) TDA literature usually includes a theoretical study of the signature's stability when proposing a new topological signature. In terms of noise in the output or weights, how robust is the R-Cross-barcode?


2) For a larger dataset, the number of simplices exponentially increases and so the construction of the VR complex would computationally be inefficient. How does your method deal with this problem?

3) A barcode is used to compare point clouds, but its superiority over other proposed metrics is unclear. Some statistical measures can also be used to compute a "divergence" or an actual metric distance between point clouds that might perform better. Mahalanobis distance between points, for instance, can be treated as instantiations of random processes.


4) Can we consider RTD as a metric?



**Summary Of The Paper:**

This paper proposes an approach for topology-preserving representation learning (dimensionality reduction). The topological similarity between data points in original and latent spaces was obtained by minimizing the Representation Topology Divergence (RTD) between original data and latent representations. They demonstrated how to make RTD differentiable and implemented it as an additional loss to the autoencoder, constructing RTD-AE. According to their computational experiments, the proposed RTD-AE better preserves the global structure of the data manifold than popular methods t-SNE and UMAP. Moreover, higher topological similarity than the alternative TopoAE method was achieved.

**Summary Of The Review:**

Overall I liked the idea of the paper and found their contribution very interesting and significant.

---

> ### Author Response · Authors · 2022-11-17
> **Response to the Review f3NA**
>
> Thank you for the positive feedback and thoughtful comments.
>
>
> **Q1**: _Theoretical study of the signature's stability. In terms of noise in the output or weights, how robust is the R-Cross-barcode?_
>
> **A**: By construction, R-Cross-barcode is the barcode of a certain weighted graph with the weights $w_1$ and $\min(w_1,w_2)$, and the robustness for the R-Cross-barcode for the point clouds follows from the robustness of the usual barcodes, and the Lipshitz continuity of the $\min(x,y)$ function. This is the basic stability  result for the R-Cross-Barcode. Next, the robustness with respect to sampling of data points follows from this, similarly to the robustness with respect to the subsampling for the usual VR barcodes as in e.g. Appendix A in [1].   Thank you for this question, we are adding this theoretical result to the text.
>
>
> **Q2**: _For a larger dataset, the number of simplices exponentially increases and so the construction of the VR complex would computationally be inefficient. How does your method deal with this problem?_
>
> **A**: Due to the use of the GPU acceleration and certain heuristics, more than 99% of simplices are rapidly eliminated from the computation.  In all our experiments on real datasets, the use of batch size equal to 256 showed good performance, while an increase of batch size often led to longer convergence without increase of quality. The computation is quite fast for batch sizes ≤ 256 since the boundary matrix is typically sparse for real datasets. We discuss this in Section 5.4.
>
> **Q3**: _Some statistical measures can also be used to compute a "divergence" or an actual metric distance between point clouds that might perform better._
>
> **A**: To answer this question, let us remark that the RTD method captures different topological aspects over multiple scales, statistical methods may fail to capture such features. An illustrative example is presented in [2] on the first page: two distinct point clouds have identical first 3 statistical moments. See also the comparison of RTD against CKA and SVCCA in [3], section 3, Figures 3,4,6,12.
>
> **Q4**: _Can we consider RTD as a metric?_
> **A**:
> * $RTD_k(X,Y) = RTD_k(Y,X)$, as we use the symmеtrized variant of RTD.
> * if $RTD_k(X, Y)=0$, for all $k\ge0$, then barcodes of $X$ and $Y$ coincide for all degrees;
> * $RTD_k(X, X)$ = 0
> * The triangle inequality for $RTD_1$ holds approximately. For one experiment with the collections of point clouds (neural representations) it was true for 97% of triplets of point clouds, see more details in [3]. Under certain conditions, the triangle inequality holds and RTD can be made into metrics, we plan to describe this in a future paper.
>
> References:
> [1] Chazal, F., Fasy, B., Lecci, F., Michel, B., Rinaldo, A., & Wasserman, L. (2015). Subsampling methods for persistent homology. ICML’2015, PMLR, p. 2143-2151.
> [2] Tsitsulin, A., Munkhoeva, M., Mottin, D., Karras, P., Bronstein, A., Oseledets, I., & Müller, E. (2019) The shape of data: intrinsic distance for data distributions.  ICLR’20.
> [3] Barannikov, S., Trofimov, I., Balabin, N., & Burnaev, E. (2021). Representation topology divergence: a method for comparing neural network representations.  ICML’22, PMLR, vol 162, 1607-1626
>
> _Concluding remarks._ Please respond to our post to let us know if the clarifications above suitably address your concerns. We are happy to address any remaining points during the discussion phase; if the responses above are sufficient, we kindly ask that you consider raising your score.

---

> > ### Comment · Reviewer_f3NA · 2022-12-07
> > **Thank you for the clarification!**
> >
> > I very much appreciate the authors' clarifications! They address my concerns very well and I am still in favour of accepting this paper. So, I would like to increase my score for the final decision.

---

### Official Review · Reviewer_8g5w · 2022-10-25

**Confidence:** 4
**Correctness:** 2
**Technical Novelty And Significance:** 2
**Empirical Novelty And Significance:** 2
**Recommendation:** 3

**Clarity, Quality, Novelty And Reproducibility:**

As indicated above, the paper would benefit from a clearer exposition to the RTD loss, especially Fig 2 is unclear to me.

The method itself looks reasonable, due to the other weaknesses I haven't checked though all the supplementary math in great detail.

In terms of novelty, one may ask of course what the benefit of an additional PH-based dim red method is -- given that there exist solutions already. Improved performance may be an argument, but I am not fully convinced that is the case here, given that we compare against default parameters (if I understand that correctly).




**Strength And Weaknesses:**

Strengths:
- Preserving the global structure in non-linear dim. red is important and underexplored
- RTD loss more closely reflects metric properties than e.g. previous TopoAE loss (but is it really a metric, or just identity of indiscernibles?)
- Many baselines considered.


Weaknesses:
- The paper lacks in clarity. Most strikingly this is evident in Fig 2, which should give a clear and simple high-level overview. However, I found it just confusing.
- The required two-step training schedule makes the method less likely to be useful in practice. In general, AE-based dim red / visualization techniques are more challenging to apply for end users out of the box (compared to UMAP, t-SNE etc), so adding an additional layer of complexity like two different training schemes makes it less likely that this method will be widely used by the community.
- Certain experimental results look a bit as if they were optimized to favor the proposed method. For instance, in Fig 1, the proposed method (panel c)) is rotated exactly as the original input data (by chance?), whereas e.g. f) that also looks reasonable is rotated differently making it look worse than it may be. Another reason why certain plots may be unfair to the baselines: You write "All of the representations were generated with default parameters of baseline methods." If this is the case, but the proposed method was tuned in a hyperparameter search, then there is no surprise that on the proposed method shines in several plots. And its also no surprise that certain methods seem to perform poorer than in their original paper (e.g. 2D visualization of Spheres with TopoAE). Another thing raising a minor red flag: The experimental structure / setup follows neatly the TopoAE paper, however certain parameters are chosen differently without explaining why. E.g. 3D visualisation of the Spheres (instead of 2), or 16D latent space instead of 2. Without any explanation, one is tempted to assume that these slightly arbitrary configurations were searched over to find best performing settings.

**Summary Of The Paper:**

The paper proposes RTD-AE, a topology-based autoencoder that aims to preserve topological information from the data space in latent embeddings. For this, RTD-AE leverages Representation Topology Divergence (RTD), a technique that allows to measure the similarity of VR complexes between two point clouds, here the data space and latent space.

The paper considers an interesting problem, that may offer some benefits over existing methods. However, there are several small-to-medium red flags that currently let me question the validity of the results and the interpretations followed from them.

**Summary Of The Review:**

Overall, the paper proposes an interesting idea, but the execution of the experiments, the clarity of the paper as well as the demonstration of the benefits seem insufficient to me at this point.

---

> ### Author Response · Authors · 2022-11-17
> **Response to the Review 8g5w**
>
> Thank you for your time and valuable feedback. We will improve the presentation according to the suggestions. Below we address specific concerns one by one.
>
> **Q1**:  _Clearer exposition to the RTD loss, especially Fig 2_
>
> **A**: Thank you for the feedback. Our method is illustrated on Figure 3. Figure 2 is from the preliminaries (background) section and illustrates results of another paper. Figure 2 shows the step-by step timeline for appearance-disappearance of topological features distinguishing the two graphs. The appearance-disappearance process is illustrated by the underlying bars, connecting the corresponding thresholds, which is the standard presentation in topological data analysis. We are adding a further clarification on this to the Figure 2 captions. To understand the representation topology divergence (RTD) definition we refer the reader to the original ICML'22 paper [1].
> We realize that Topological Data Analysis is not well known in the ML/AI community.
> For the reader's convenience, we included more details on Topological Data Analysis, please see Appendix A, and the formal definition of RTD, please see Appendix B.
>
> **Q2**: _Two-step training schedule makes the method less likely to be useful_
>
> **A**: Initial training with the reconstruction loss only is not an essential part of the proposed method; it is for speedup only.
> We note that popular methods also employ two-step procedures. For example, t-SNE uses “early_exaggeration”, which means that for the first 50 epochs datapoint similarities $p_{ij}$ are multiplied by 4 to achieve better clusterization. In UMAP, representations are initialized with the eigenvectors of the normalized graph Laplacian. These procedures work “under the hood” and do not hinder wide application of t-SNE and UMAP in the community.
>
> **Q3**: _Certain experimental results look a bit as if they were optimized to favor the proposed method. For instance, in Fig 1, the proposed method (panel c)) is rotated exactly as the original input data (by chance?), whereas e.g. f) that also looks reasonable is rotated differently making it look worse than it may be._
>
> **A**: For all the methods (AE, RTD-AE, TopoAE, t-SNE, UMAP) in Figure 1 we tried vertical and horizontal flips and swapping of X and Y axis to align 2D representations with 3D original data as much as possible.
> We note that for t-SNE and UMAP the alignment is not the major problem, but the total incoherence in global and local structure; rotating either of these two figures does not change this conclusion.
> See also the results of PacMAP and IVIS in Appendix N, which also perform poorly. Notice that RTD-AE performs better than TopoAE (panel f)) in terms of the four quality measures (Table 5, Appendix G) which are rotation-invariant.
>
>
> **Q4**: _You write "All of the representations were generated with default parameters of baseline methods." If this is the case, but the proposed method was tuned in a hyperparameter search, then there is no surprise that on the proposed method shines in several plots._
>
> **A**: For RTD-AE, we didn't perform a hyperparameter search on a grid or anything like this. The only hyperparameters we have are the generic ones like learning rate or width/depth of a neural network. We selected them by common deep learning principles. Architectures and initializations of neural networks were the same for all autoencoder-based methods (AE, TopoAE, RTD-AE). The proposed method, RTD-AE, doesn't even have a mixing coefficient in front of the topological loss (the coefficient in front of the topological loss always equals 1), in contrast with TopoAE.
> We consider that the ability of RTD-AE to perform well on the variety of datasets without tuning of hyperparameters is an indicator of its robustness.
> To further validate the performance of RTD-AE, we have made additional experiments with complete hyperparameters searching (learning rate, batch size and lambda, as in [2]) for the TopoAE on the COIL-20 dataset:
>
> |   | L.C. | W.D.H0   |  T.A.  | RTD   |
> |---|:---:|:---:|:---:|:---:|
> |  TopoAE (tuned) | 0.821  |161.7   |0.793   | 6.72   |
> | RTD-AE (default hyperpar) |  **0.944** | **88.9**   | **0.892**  | **5.78**   |
>
> The results of RTD-AE (without hyperparameters searching) are still better than TopoAE for all four quality measures.
>
>
> _(Q-A response continues below, please see the next comment for Q5-Q7 and the references)_

---

> > ### Author Response · Authors · 2022-11-17
> > **Response to the Review 8g5w (continued, please see Q1-Q4 in the above comment)**
> >
> > **Q5**: _Certain methods seem to perform poorer than in their original paper (e.g. 2D visualization of Spheres with TopoAE)._
> >
> > **A**: In the original TopoAE paper [2], authors did a complete search of hyperparameters, over a total of 3 hyperparameters for TopoAE, and presented the best configuration. This is the  source of the difference in the two visualizations of the TopoAE 2D Spheres results. We have performed an additional experiment using the grid search to reproduce the 2D visualization of Spheres as close as possible to the original paper [2] and to compare it with RTD-AE.
> > After the grid search, the RTD-AE still performed better than TopoAE in the linear correlation, Wasserstein Distance in H0 and RTD quality measures:
> >
> > |   |  L.C.&uarr;|  W.D.H0&darr;| T.A.&uarr;|  RTD&darr;|
> > |---|:---:|:---:|:---:|:---:|
> > |TopoAE (tuned)   | 0.691  |  43.298±1.629 | 0.3688±0.0165  |  39.837±1.318 |
> > |RTD-AE (tuned)   | **0.706**  |  **42.133±1.683** | **0.3765±0.0124**  | **37.286±1.393**  |
> >
> > For TopoAE and RTD-AE the difference between means in WDH0 and TA is lesser than the standard derivations, and due to this, we have performed the one-tailed Student's t-test to verify their relation. According to the t-test's results, we can reject the null hypothesis that the mean WDH0 for TopoAE is lower than the mean WDH0 for RTD-AE at the significance level of 0.05. Same result confirming the better performance of RTD-AE was obtained for the triplet accuracy.
> >
> > **Q6**: _The experimental structure / setup follows neatly the TopoAE paper, however certain parameters are chosen differently without explaining why. E.g. 3D visualisation of the Spheres (instead of 2), or 16D latent space instead of 2_.
> >
> > **A**:
> > * We have presented experiments with a lot more different datasets and setups, if compared to this paper. In total there are 6 synthetic datasets in our paper, in comparison with 1 synthetic dataset in the TopoAE paper. We have used 4 more different real-world datasets, in comparison with the TopoAE paper, the extra datasets include scRNA mice, scRNA melanoma, Mammoth, COIL-20.
> >
> > * The 2-D visualizations of the Spheres and  their quality measures  are presented in Figure 11 and in Table 9 in Appendix L. The RTD-AE performs better than TopoAE, either with default hyperparameters or after the grid search (see the table above).
> >
> > * We have presented the results for 2D latent space in Table 5,  Appendix D. Our method still performs well w.r.t. baselines, but with a smaller margin. We mention it in Section 5.2. The reason we preferred 16D space to 2D is that it’s impossible to preserve topological structure of real high-dimensional data in 2D space without distortions.
> >
> > **Q7**: _What the benefit of an additional PH-based dim red method is._
> >
> > **A**:
> > * Minimization of the RTD loss guarantees coincidence of topological features and their localizations for H0 and for higher dimensions (1d-cycles, tunnels, k-voids etc). At the same time, the topological loss from TopoAE doesn’t have such guarantees and can be discontinuous in rather standard situations, please see the example in Appendix&#160;J.
> >
> > * The TopoAE paper was a valuable achievement that incorporated a topological loss term into the autoencoder setting, however its performance is far from ideal: in particular, because it lacks continuity,  to get satisfactory results, the TopoAE requires the fine-tuning of hyperparameters via the grid search, contrary to the proposed method.
> >
> > * The described framework for optimization of the RTD loss permits optimizing a data representation so that  its multiscale k-dimensional topological features (clusters, 1-cycles, k-voids etc) become similar to  that of a given data representation for any dimension k≥0, bypassing the standard scheme comparing two PH-diagrams.
> >
> > References:
> > [1] Barannikov, S., Trofimov, I., Balabin, N., & Burnaev, E. (2021). Representation Topology Divergence: A Method for Comparing Neural Network Representations. In ICML’22. PMLR, vol 162, 1607-1626
> > [2] Moor, M., Horn, M., Rieck, B., & Borgwardt, K. (2019). Topological Autoencoders. In International Conference on Machine Learning. PMLR vol 119, 7045-7054.
> >
> > _Concluding remarks._ Please respond to our post to let us know if the clarifications above suitably address your concerns. We are happy to address any remaining points during the discussion phase; if the responses above are sufficient, we kindly ask that you consider raising your score.

---

### Decision · Program_Chairs · 2023-01-20

**Decision:**

Accept: poster

**Justification For Why Not Higher Score:**

N/A

**Justification For Why Not Lower Score:**

N/A

**Metareview: Summary, Strengths And Weaknesses:**

This paper proposes an approach for topology-preserving representation learning (dimensionality reduction). The topological similarity between data points in original and latent spaces was obtained by minimizing the Representation Topology Divergence (RTD) between original data and latent representations. The proposed RTD-AE better preserves the global structure of the data manifold than popular methods t-SNE and UMAP.

This paper received divergent scores before the authors' rebuttal. And it looks like the authors did a great job addressing reviewers' concerns, and three out of four reviewers have increased their scores to 8. The meta-reviewer agrees with the positive reviewers that this paper provides a general justification and explanation of how to differentiate and optimize topology-based functions, which would be of wide interest to the community. The meta-reviewer recommends acceptance.

**Note From Pc:**

if the above contains the word "oral" or "spotlight" please see: "oral" presentation means -> notable-top-5% and "spotlight" means -> notable-top-25%. As stated in our emails, we are disassociating presentation type from AC recommendations

**Summary Of Ac-Reviewer Meeting:**

Three reviewers increased their scores to 8 and one reviewer didn't respond.